# Modest volcanic SO$_2$ emissions from the Indonesian archipelago

Philipson Bani [1,2✉], Clive Oppenheimer [3], Vitchko Tsanev[3], Bruno Scaillet [4], Sofyan Primulyana[5], Ugan Boyson Saing[5], Hilma Alfianti[5] & Mita Marlia[5]

Indonesia hosts the largest number of active volcanoes, several of which are renowned for climate-changing historical eruptions. This pedigree might suggest a substantial fraction of global volcanic sulfur emissions from Indonesia and are intrinsically driven by sulfur-rich magmas. However, a paucity of observations has hampered evaluation of these points—many volcanoes have hitherto not been subject to emissions measurements. Here we report new gas measurements from Indonesian volcanoes. The combined SO$_2$ output amounts to 1.15 ± 0.48 Tg/yr. We estimate an additional time-averaged SO$_2$ yield of 0.12-0.54 Tg/yr for explosive eruptions, indicating a total SO$_2$ inventory of 1.27-1.69 Tg/yr for Indonesian. This is comparatively modest—individual volcanoes such as Etna have sustained higher fluxes. To understand this paradox, we compare the geodynamic, petrologic, magma dynamical and shallow magmatic-hydrothermal processes that influence the sulfur transfer to the atmosphere. Results reinforce the idea that sulfur-rich eruptions reflect long-term accumulation of volatiles in the reservoirs.

[1] Laboratoire Magmas et Volcans, Université Blaise Pascal-CNRS-IRD, OPGC, 63170 Aubière, France. [2] Centre IRD de la Nouvelle-Calédonie, 101, Promenade Roger Laroque, BP A5, 98848 Nouméa Cedex, Nouvelle-Calédonie, France. [3] Department of Geography, University of Cambridge, Downing Place, Cambridge CB2 3EN, UK. [4] Institut des sciences de la Terre d'Orléans, Université d'Orléans-CNRS-BRGM, 1a rue de la Férollerie, 45071 Orléans, France. [5] Center for Volcanology and Geological Hazard Mitigation, Jl. Diponegoro No. 57, Bandung 40122, Indonesia. ✉email: philipson.bani@ird.fr

While not the most abundant species in volcanic gases, sulfur dioxide ($SO_2$) is the easiest to measure remotely with the aim of deriving a flux. This owes principally to its absorption of ultraviolet (UV) light, enabling daytime spectroscopic measurements from the ground, air and space[1]. Measurements of $SO_2$ flux are a cornerstone of volcano monitoring and contribute to the understanding volcanic degassing. They permit the calculation of fluxes of other volcanic gas species ($X$) from measurements of their ratios to sulfur dioxide ($X/SO_2$) and underpin global inventories of volcanic gas emissions to the atmosphere. In this respect, $SO_2$ is particularly important given its roles in atmospheric chemistry and radiation[2].

Advances in satellite remote sensing of $SO_2$ in both ultraviolet and infrared wavebands are adding to our knowledge[3,4] and the proliferation of compact UV spectrometers and cameras[5–7], is enabling measurements at less accessible volcanoes. However, the compilation of global inventories still faces numerous challenges, including temporal and spatial data gaps, measurement uncertainties, the presence of multiple sulfur species (including S, $H_2S$ and $H_2SO_4$) in volcanic emissions, and the challenges of processing large volumes of data.

One notable lacuna in $SO_2$ inventories is the Indonesian archipelago. According to Siebert et al.[8], there are 78 historically active volcanoes in Indonesia, i.e., those with at least one historically-recorded eruption. But such a definition finds its limit in Indonesia where the documentary record is incomplete and traditional knowledge lost or not fully integrated into scientific records. A more complete inventory of the Indonesian volcanoes can be found in the "Badan Geologi" database[9] which lists 126 active volcanoes (including six submarine edifices). They are subdivided into 77 type-A volcanoes, which have experienced at least one increase in magmatic and/or phreatic activity since 1600; 29 type-B volcanoes with solfataric and/or fumarolic manifestations but no eruption since 1600; and 20 type-C, which are solfataras and/or fumarole fields lacking a defined volcanic edifice (Fig. 1 and Table 2). These volcanoes fall within four distinct arcs: Sunda, Banda, Sangihe, and Halmahara (Fig. 1).

In the first global compilations of the volcanic $SO_2$ budget, the Indonesian contribution was unspecified[10]. Subsequently, over four decades, new observations have furnished estimates of annual $SO_2$ inventories for Indonesia (Table 1). These have varied considerably, beginning with the work of Le Guern[11], who estimated 0.07 Tg /yr (representing just 0.15% of the global volcanic $SO_2$ budget) compared with a more recent estimate of Carn et al.[3] of 2.2 Tg $SO_2$ /yr (representing 9.5% of the global volcanic budget). Note that these and intervening works have also reported disparate figures for the global total (Table 1), which is unsurprising given the different datasets (and their timespans) and methods employed. Despite developments in $SO_2$ sensing, hitherto only a fraction of the more than 100 Indonesian volcanoes classified as active has been subject to $SO_2$ flux measurement campaigns.

Here we present a new inventory of volcanic $SO_2$ emissions for Indonesia based on portable ground-based remote sensing instruments and systematic program of fieldwork observations. We focus our efforts on the subaerial type-A volcanoes with passive degassing, which we consider based on field observation to be the main volcanic degassing sources in Indonesia (Table 2 and Fig. 1). We use the term 'passive' to refer to the style of gas emission so as to distinguish it from larger, sporadic explosive emissions, though the term can encompass a wide range of sources from magmatic to fumarolic. We evaluate the factors influencing the variations identified between volcanoes and between sub-regions of Indonesia and consider the total $SO_2$ emission rate for the archipelago in the global context.

## Results

**$SO_2$ emission budget.** Of the 73 aerial type-A volcanoes across Indonesia, we conducted measurements at 47 (Fig. 1 and Table 3), including 12 that exhibit negligible $SO_2$ release. Of the remaining 26 volcanoes that were not visited, 20 are either inactive or exhibit negligible $SO_2$ emission, according to local observatory reports and available data and images (https://vsi.esdm.go.id/). There were six volcanoes known for persistent degassing that we did not visit: Banda Api, Serua, Batu Tara, Sangeang Api, Rinjani, and Arjuno Welirang (Fig. 2). However, satellite observations provide some constraints for Batu Tara, Rinjani and Sangeang Api[3].

The Sunda arc makes the largest contribution, with a collective daily output of $1313 \pm 539$ Mg (Fig. 2). Sinabung, Kawah Ijen, Slamet, Anak Krakatau and Bromo volcanoes are the strongest $SO_2$ sources of the arc at $275 \pm 25$ Mg/d, $238 \pm 194$ Mg/d, $206 \pm 66$ Mg/d, $190 \pm 77$ Mg/d, and $166 \pm 2$ Mg/d, respectively. Moderate to small $SO_2$ emission rates have been reported for Rinjani ($74 \pm 65$ Mg/d), Sangeang Api ($71 \pm 75$ Mg/d)[3], Semeru ($48 \pm 22$ Mg/d)[12], Merapi ($20 \pm 7$ Mg/d), Kerinci ($9.8 \pm 4.3$ Mg/d) and Kaba ($9.0 \pm 3.1$ Mg/d) (Table 3). These 11 volcanoes are the main degassing sources of the Sunda arc. Measurements have also quantified minor to negligible $SO_2$ emission rates (0.2–2.6 Mg/d, Table 3) for five other volcanoes, namely Marapi, Tangkuban Parahu, Papandayan, Talang and Guntur. In total there are 16 volcanic $SO_2$ degassing sources across the Sunda arc. Two of them, Dempo and Kelut, host crater lakes that trap condensable gases, limiting their atmospheric contribution. The remaining 19 type A volcanoes out of a total 37 volcanoes across the Sunda are either quiescent (non-emitters) or characterized by low temperature solfataras and/or fumaroles (Table 3), except Ajurno Welirang, which sustains a persistent but minor degassing.

The Banda arc has the lowest passive $SO_2$ degassing budget with a total daily output of $330 \pm 175$ Mg. Ili Lewotolo, Sirung, and Lewotobi Perempuan are the main sources but with moderate emission rates corresponding to $75 \pm 40$ Mg/d, $48 \pm 22$ Mg/d, $15 \pm 10$ Mg/d. Batu Tara and Rokatenda with $102 \pm 51$ and 60 Mg/d, respectively[3], are among the main $SO_2$ degassing sources of the Banda arc. Note that the figure for Batu Tara was obtained by subtracting our measurement for Ili Lewotolo from the reported combined flux for both volcanoes[3]. Lesser $SO_2$ emission rates are found for Iya ($8 \pm 6$ Mg/d), Wurlali, ($8 \pm 6$ Mg/d) and Ebulobo ($6 \pm 3$ Mg/d). We obtain negligible fluxes for Egon ($3 \pm 2$ Mg/d), Kelimutu ($2.0 \pm 0.7$ Mg/d), Lewotobi Lakilaki ($2.0 \pm 0.7$ Mg/d), and Ili Werung ($1.0 \pm 0.8$ Mg/d) (Table 3). Two other volcanoes of the Banda arc, including Serua, and Banda Api, were not visited and therefore their $SO_2$ emissions remain unknown. However, based on information from the local observatories, degassing strength of Banda Api is comparable to that of Wurlali, and exceeds that of Serua. The lack of measurements is thus unlikely to bias significantly our arc-scale flux estimate. The $SO_2$ degassing associated with low temperature solftaras and fumaroles from the remaining eight type A volcanoes of this arc is negligible (Table 3).

The Sangihe arc hosts three volcanoes with relatively strong $SO_2$ emission rates, including Soputan ($376 \pm 100$ Mg/d), Lokon ($117 \pm 10$ Mg/d) and Karangetang ($120 \pm 55$ Mg/d). Awu the northernmost volcano of the arc emits $13 \pm 5$ Mg $SO_2$/d. The $SO_2$ contribution from the five remaining volcanoes is negligible (Table 4). Hence, with a total $SO_2$ degassing budget of $626 \pm 170$ Mg per day, the Sangihe arc constitutes a notable arc-scale volcanic degassing source to the atmosphere. Note, however, that Soputan clearly stands out as the strongest source, representing 60% of the total for this arc.

Finally, the $SO_2$ emission rate from the Halmahera arc amounts to $897 \pm 437$ Mg/d with more than 90% of this flux

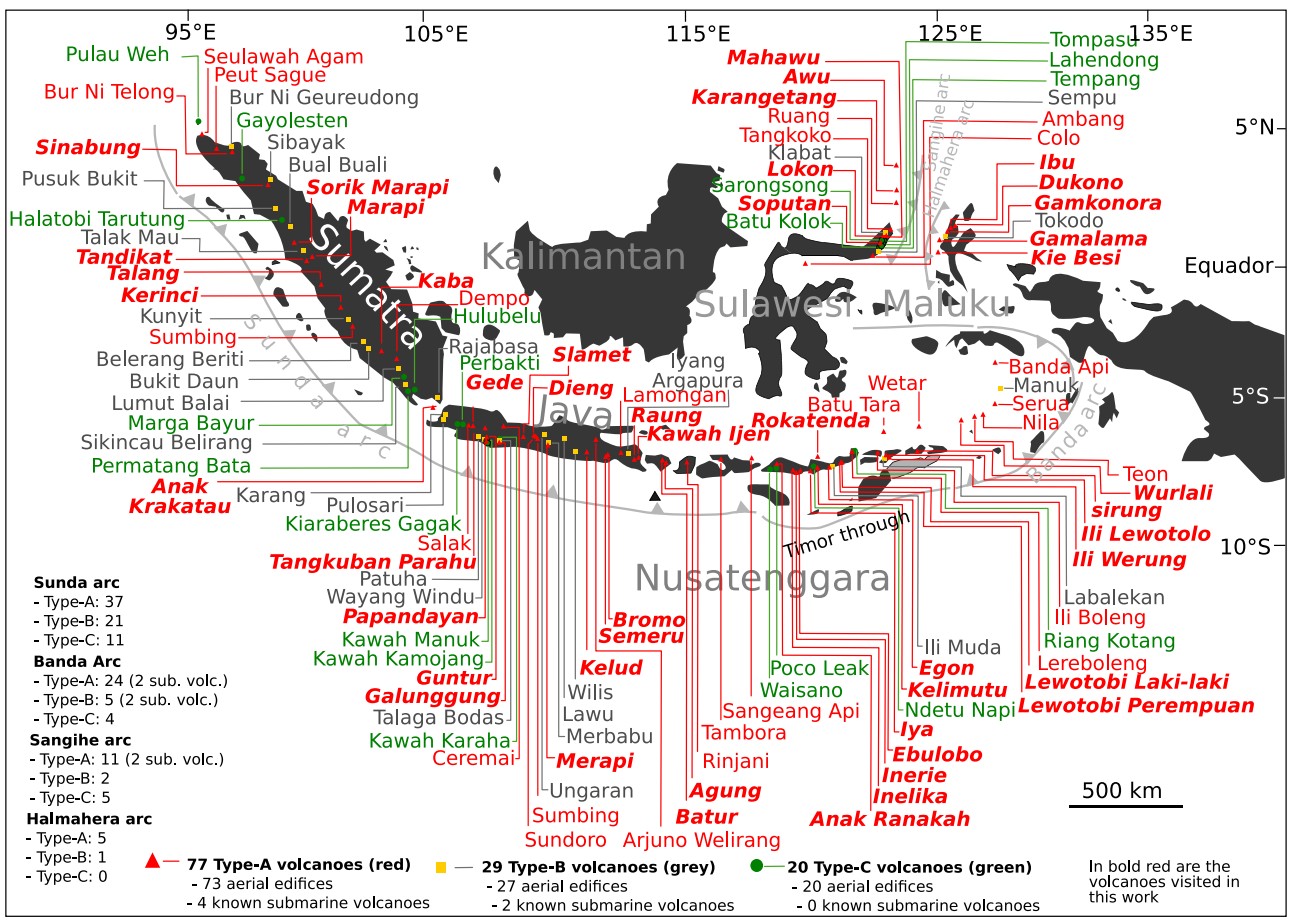

**Fig. 1 Indonesian active volcanoes.** The distribution of the 126 active volcanoes across the archipelago of Indonesia, including 120 aerial and six known submarine edifices (not shown on the map). 77 are classified as Type-A (red triangles), 29 as type-B (yellow squares) and 20 as type-C (green circles) The volcanoes visited in this work are highlighted in red-bold-italic.

accounted for by Dukono which emits $819 \pm 394$ Mg/d. The other volcanoes are low to moderate sources, with $59 \pm 32$ Mg/d from Ibu, $16 \pm 10$ Mg/d from Gamalama and $3.4 \pm 1.0$ Mg/d from Gamkonora. On Kie Besi, only a small fumarole is present and its $SO_2$ emission is considered negligible.

Based on the above results, and as summarized in Table 3, the total daily $SO_2$ emission passively released into the atmosphere from the entire Indonesian archipelago is $3200 \pm 1300$ Mg/d, equivalent to $1.15 \pm 0.48$ Tg $SO_2$ yr$^{-1}$ (Table 3). We emphasise that this figure is representative of the periods of observations and must be viewed cautiously but we believe it gives a useful guide to the scale of emissions at the scale of the entire archipelago.

**Principal point sources**. Our ranking of Indonesian volcanic sources of passive degassing is shown in Table 3. Dukono (Halmahera arc) is the strongest, representing more than a quarter (26%) of the total. Soputan (Sangihe arc), Sinabung and Kawah Ijen (Sunda arc) are also notable, representing, 12%, 9% and 8% of the total, respectively. These four volcanoes alone constitute around half of the total inventory. They are sustained by different magma compositions, i.e., basaltic andesite to andesite on Sinabung[13], basaltic to dacite on Kawah Ijen, basaltic on Soputan[14], and andesite to trachyandesite on Dukono[15]. Six other volcanoes exhibit moderate rates of $SO_2$ emission, including Slamet (206 Mg/d), Anak Krakatau (190 Mg/d), Bromo (166 Mg/d), Karangetang (120 Mg/d), Lokon (117 Mg/d) and Batu Tara (102 Mg/d). Thus, ten volcanoes contribute 82% of the total passive volcanic $SO_2$ emission budget of Indonesia. There are five other volcanoes with modest $SO_2$ fluxes,

between 50 and 100 Mg/d that together represent 11% of the budget, seven with $SO_2$ emission rates between 10 and 50 Mg/d, representing 5% of the budget, and finally 14 volcanoes whose $SO_2$ degassing is below 10 Mg/d.

**Arc scale variations**. Our new $SO_2$ inventory reveals substantial variations in $SO_2$ output between the arcs of the Indonesian archipelago. The 3000-km-long Sunda arc, with 37 type A volcanoes, is the largest $SO_2$ source at $0.48 \pm 0.20$ Tg/yr representing 41% of the Indonesian total. The 2000-km-long Banda arc, in contrast, contributes just $0.12 \pm 0.06$ Tg/yr representing only 10% of the total, despite hosting 24 Type A volcanoes. The 600-km-long Sangihe and 500-km-long Halmahera arcs are stronger sources ($0.23 \pm 0.06$ Tg and $0.33 \pm 0.16$ Tg $SO_2$ yr$^{-1}$, respectively) despite their shorter extents and comparatively few volcanoes (eleven and five, respectively). It is thus plausible that the geodynamic contexts play a key role in the $SO_2$ emission budget of Indonesia; in terms of $SO_2$ emission rate per km of arc, the Halmahera arc is the strongest source with an output of 655 Mg $SO_2$ per km yr$^{-1}$ followed by the Sangihe, Sunda and Banda arcs with 380, 160 and 60 Mg $SO_2$ per km yr$^{-1}$, respectively.

**Passive and explosive degassing**. Our total Indonesian $SO_2$ inventory of $1.15 \pm 0.48$ Tg/yr based on ground-based and airborne surveys of persistent volcanic degassing across the archipelago is half the estimated emission from the 20 Indonesian volcanoes reported in ref. [3], derived from satellite remote sensing measurements. But this latter approach focused on a different

**Table 1 Reported global volcanic SO$_2$ inventories and contribution from Indonesian volcanoes.**

| Authors | Total volcanic SO$_2$ (Tg/yr) | Method(s) | Contribution from Indonesian volcanoes |
|---|---|---|---|
| Le Guern, 1982[11] | 50.0 | Using Correlation Spectrometer (COSPEC) data from ref. [50], and extrapolating to a larger number of active volcanoes in different geodynamic provinces. | 0.073 Tg yr$^{-1}$ from Indonesia (Merapi volcano was considered as the main degassing source). |
| Spiro et al., 1992[51] | 19.2 | Based on plume size, following ref. [52] and referring to volcanism in 1964–1972 and 1980: 28% of the SO$_2$ emission budget from passive degassing, assuming 61% from eruption to troposphere and 11% to the stratosphere. | 0.41 Tg yr$^{-1}$ attributed to Indonesian volcanoes. |
| Andres and Kasgnoc, 1998[53] | 13.4 | From a compilation of S fluxes in 214 published references, personal comm., and conference presentations. Electronic mail messages were sent to the VOLCANO list for data discussion with volcanologists and atmospheric scientists. Two categories were distinguished: continuously (49) and sporadically erupting (25) volcanoes. | Indonesian volcanoes contributed to only 0.10 Tg yr$^{-1}$ (four volcanoes were considered, including Merapi, Tangkuban parahu, Bromo and Slamet). |
| Halmer et al., 2002[54] | 15.0–21.0 | Considering the SO$_2$ emissions of 50 volcanoes recorded by the Total Ozone Mapping Spectrometer (TOMS) and COSPEC, then extrapolated to 310 unmeasured volcanoes based on the VEI-SO$_2$ relationship, magma composition, tectonic setting and the state of activity. | 2.1–2.6 Tg yr$^{-1}$ attributed to Indonesian subduction zone. |
| Shinohara, 2013[55] | 19.8 | Based on a literature review: 76 persistently degassing volcanoes release an estimated 18.5 Tg/yr of SO$_2$ and the time-averaged annual SO$_2$ flux from explosive eruption (1.3 Tg/yr) is obtained based on VEI-SO$_2$ emission correspondence. | 0.1 Tg from Indonesia, four volcanoes were considered: Merapi, Tangkubanparahu, Slamet and Bromo. |
| Carn et al., 2017[3] | 23.0 | Based on OMI data spanning 2005–2015 and focused on passive degassing from 91 volcanoes worldwide. | 2.2 Tg from Indonesia. 20 volcanoes were considered: Dukono, Bromo-Semeru, Lewotolo-Batu Tara, Ijen-Raung, Sirung, Sinabung, Karangetang, Krakatau, Kerinci, Slamet, Lokon, Ebulobo, Rinjani, Sangeang api, Paluweh Marapi and Merapi (from the highest to the lowest SO$_2$ emission). |

study period (2005–2015) and integrated the SO$_2$ contribution from explosive events. Between 2010 and 2020, there were 110 eruptive episodes across the Indonesian archipelago reported in the Global Volcanism Program (https://volcano.si.edu/) and Bandan Geologi (https://vsi.esdm.go.id/). Most of these were minor to moderate in scale (VEI < 3) and their SO$_2$ emissions were, mostly, not captured by satellite sensors. The year 2014 was the most active year with 14 eruptions while in 2020 there were only six eruptive events reported. The mean value is ten eruptions per year (Table 4 and Fig. 3). The Sunda arc dominates this list with 60 eruptions at 16 different volcanoes. Sinabung and Anak Krakatau were the most active with nine and eight events, respectively. Kerinci and Marapi were also notably active with, respectively, six and seven eruptions, whilst Merapi, Semeru, and Sangeang Api experienced four eruptions each. The Banda arc produced 13 eruptions at seven different volcanoes over the last decade. Batu Tara was the most active with four eruptions. For the Sangihe arc, 19 eruptions were reported at three different volcanoes, including Karangetang, the most active with ten eruptions, Soputan with six eruptions and Lokon with three eruptions. Eighteen eruptive events were recorded for the Halmahera arc. Dukono was the most active volcano with 11 eruptions, followed by Gamalama with six eruptions.

To estimate the SO$_2$ contribution from these explosive events, we first used a formulation[16] relating volcanic explosivity index (VEI) and SO$_2$ yield:

$$\log 10(SO_2, Tg) = 0.71 VEI - 3.15 \qquad (1)$$

We took VEI values reported in the Global Volcanism Program (https://volcano.si.edu/). This indicates a total eruptive SO$_2$

output over the 2010–2020 period of 5.99 ± 0.31 Tg with annual totals varying between 0.25 Tg and 1.03 Tg (Table 4, Fig. 3). The Sunda arc released 3.7 ± 0.3 Tg, representing 62% of the total, with the main contributions from Sinabung, Merapi, Kelut, Anak Krakatau, and Agung. For the Banda arc we estimate 0.40 ± 0.05 Tg SO$_2$ (0.04 Tg/yr) accounting for 7% of the budget, mostly contributed by Rokatenda, Lewotolo and Batu Tara. For the Sangihe arc, Soputan, Karangetang and Lokon volcanoes were the only sources with an estimated combined yield of 0.75 ± 0.05 Tg (0.07 Tg/yr) of SO$_2$ or 12% of the total. Lastly, the Halmahera arc released 1.12 ± 0.01 Tg (0.10 Tg/yr) of SO$_2$ through eruptions, representing 19% of the total. Dukono was the main contributor, accounting for 83% of the arc's output (Table 4).

We also analyzed available satellite data (http://SO2.gsfc.nasa.gov/) for the SO$_2$ mass over Indonesia between 2010 and 2020. Out of the 110 eruptions, 71 were captured by the satellite (64%) and the corresponding SO$_2$ mass amounts to a total of 1.31 ± 0.18 Tg, with a mean annual value of 0.12 ± 0.04 Tg. The Sunda and Halmahera arcs are the main contributors representing, respectively, 81% (1.07 ± 0.18 Tg) and 11% (0.14 ± 0.01 Tg) of the total SO$_2$. The main contributors from the Sunda arc are Kelut (44%), owing to its 2014 eruption[17], Merapi (14%), primarily related to its 2010 event[18], and Sinabung (12%), which has experienced episodic dome growth and collapse since 2010 (ref. [19]). For the Halmahera arc, 99% of the arc contribution is from Dukono, reflecting its continuous eruptive activity. For the Sangihe arc, the total SO$_2$ yield amounts to 0.03 ± 0.005 Tg, and Soputan, with its recurrent eruptive activity, is the main source, representing 67% of the arc contribution. Finally, the Banda arc contribution corresponds to 0.068 ± 0.019 Tg and is dominated by the 2020

**Table 2 The distribution of the Indonesian active volcanoes per type (A, B or C), region and arcs.**

| | Type-A | Type-B | Type-C | | Type-A | Type-B | Type-C |
|---|---|---|---|---|---|---|---|
| Sunda Arc | **Sumatra** | | | Banda Arc | **Flores-Lembata-Pantar** | | |
| | Seulawah Agam | Bur Ni Geureudong | Pulau Weh | | Anak Ranaka | Ili Muda | Waisano |
| | Peut Sague | Sibayak | Gayolesten | | Inelika | Labalekan | Poco Leak |
| | Bur Ni Telong | Pusuk Bukit | Halatobi Tarutung | | Inerie | Yersey (sub. volc.) | Ndetu Napi |
| | Sinabung | Bual Buali | Halubelu | | Ebulobo | | Riang Kotang |
| | Sorik Marapi | Talak Mau | Marga Bayur | | Iya | | |
| | Tadikat | Kunyit | Permatang Bata | | Kelimutu (Paluweh) | | |
| | Marapi | Bemerang Beriti | | | Rokatenda | | |
| | Talang | Bukit Daun | | | Batu Tara | | |
| | Kirinci | Lumut Balai | | | Egon | | |
| | Sumbing | Sikicau Belirang | | | Lewotobi Laki-Laki | | |
| | Kaba | Rajabasa | | | Lewotobi Perempuan | | |
| | Dempo | | | | Lereboleng | | |
| | Anak Krakatau | | | | Ili Boleng | | |
| | **Java** | | | | Ili Werung | | |
| | Salak | Karang | Kiaraberes Gagak | | Ili Lewotolo | | |
| | Gede | Pulosar | Perbakti | | Sirung | | |
| | Tangkuban Parahu | Patuha | Kawah Manuk | | Hobal (sub. volc.) | | |
| | Papandayan | Wayang Windu | Kawah Kamojang | | **South Maluku** | | |
| | Guntur | Talaga Bodas | Kawah Karaha | | Wetar | Manuk | |
| | Galunggung | Ungaran | | | Nieuwerkerk (sub. volc.) | Emp. China (sub. volc.) | |
| | Ceremai | Merbabu | | | Wurlali | | |
| | Slamet | Lawu | | | Teon | | |
| | Dieng | Wilis | | | Nila | | |
| | Sundoro | Lyang Argapura | | | Serua | | |
| | Sumbing | | | | Banda Api | | |
| | Merapi | | | Halmahera Arc | **North Maluku** | | |
| | Kelut | | | | Kie Besi (Makian) | Tokodo | |
| | Ajurno Welirang | | | | Gamalama | | |
| | Semeru | | | | Gamkonora | | |
| | Bromo | | | | Ibu | | |
| | Lamongan | | | | Dukono | | |
| | Raung | | | Sangihe Arc | **Sangihe** | | |
| | Kawah Ijen | | | | Colo | Sempu | Tempang |
| | **Bali-Lombok-Sumbawa** | | | | Ambang | Klabat | Lahendong |
| | Batur | | | | Mahawu | | Tompasu |
| | Agung | | | | Soputan | | Batu Kolok |
| | Rinjani | | | | Lokon | | Sarongsong |
| | Tambora | | | | Tangkoko | | |
| | Sangeang Api | | | | Ruang | | |
| | | | | | Karangetang | | |
| | | | | | Banua Wuhu (sub. volc.) | | |
| | | | | | Submarine 1922 (sub. volc.) | | |
| | | | | | Awu | | |

eruption of Ili Lewotolo, representing 92% of the arc contribution. These figures obtained from satellite data are lower than those calculated from the VEIs (Fig. 3 and Table 4), except for Kelut and Ili Lewotolo, where the estimates converge. Despite the discrepancy, particularly in the lower $SO_2$ estimates, both approaches highlight the Sunda and Halmahera arcs as the main $SO_2$ contributors to explosive emissions.

Combining the figures we derive for passive and explosive $SO_2$ degassing yields a total source between 1.27 and 1.69 Tg $SO_2$ /yr for the Indonesia archipelago. The lower and the higher range integrate respectively the mean annual figures from satellite data and from the VEIs (Table 4). About 10% to 30% of total $SO_2$ release was sustained by larger, sporadic explosive emissions. This Indonesia total budget corresponds to 3–7% of the global volcanic $SO_2$ emission budget based on estimates in ref. [3] and [20] (23–33 Tg/yr) and is comparable to the total emissions from Japan [21]

(Fig. 4), although much less when the degassing budget is normalized by arc length.

## Discussion

This work constitutes the first near-comprehensive $SO_2$ emission survey across the Indonesian archipelago. We estimate a passive degassing flux of 1.15 Tg $SO_2$/yr for Indonesia in the period of 2010–2019. This represents the cumulative emission from twenty volcanoes, the four strongest sources being Dukono, Soputan, Sinabung and Kawah Ijen, which together represent 54% of the total emission budget, while Slamet, Anak Krakatau, Bromo, Karangetang, and Lokon account for another 25% of the total. More modest sources include Batu Tara, Ili Lewotolo, Rinjani, Sangeang Api, Rokatenda, and Ibu, representing 14% of the total. Seven minor sources, Semeru, Slamet, Merapi, Gamalama,

**Table 3 Combined SO₂ flux results for the Indonesian volcanoes.**

| Volcano Name | long/lat | Rank | Typical degassing status | Average plume height (m) | Mean SO₂ flux (Mg/d) | error (Mg/d) | Method/ Technique | Source/ (measurement date) |
|---|---|---|---|---|---|---|---|---|
| **Sunda Arc** | | | | | | | | |
| Sinabung | 98.392E/ 3.170 N | 3 | emission from lava dome | 2600–2800 | 275 | 25 | DOAS scanner (NOVAC) | Primulyana et al. 2018 (2010–2016) |
| Marapi | 100.474E/ 0.380 S | 29 | Minor degassing from the main central crater | 2800–2900 | 2.6 | 1.5 | DOAS Walking traverses | This work (25/10/2014) |
| Talang | 100.681E/ 0.979 S | 35 | Minor degassing from fumaroles formed along a NE-SW fracture transecting the summit | 2400–2500 | 0.3 | 0.04 | DOAS Walking traverses | This work (27/06/2012) |
| Kerinci | 101.264E/ 1.697 S | 22 | Emission from a lava dome in the crater and frequent eruptions | 3200–4300 | 9.8 | 4.3 | DOAS Walking traverses | This work (03/05/2012) |
| Kaba | 102.615E/ 3.522 S | 23 | Degassing of the extended fumarole zone in the crater | 1700–1800 | 9.0 | 3.1 | DOAS Walking traverses | This work (02/06/2015) |
| Anak Krakatau | 105.423E/ 6.102 S | 6 | Outgassing through the summit crater filled by a lava flow | 400–500 | 190 | 77 | DOAS Airborne traverses | This work (02/04/2013) |
| Tangkuban Parahu | 107.600E/ 6.770 S | 32 | Minor gas release from the main crater | 1850–1900 | 1.8 | 0.4 | DOAS Walking traverses | This work (06/09/2012) |
| Papandayan | 107.730E/ 7.320 S | 33 | Hydrothermal dominated gas through three primary fumarolic zones | 2250–2300 | 1.4 | 0.8 | DOAS Walking traverses | Bani et al. 2013 (18/06/2011) |
| Guntur | 107.841E/ 7.143 S | 36 | Gas emissions from solfatara | 2100–2150 | 0.2 | 0.1 | DOAS Walking traverses | This work (21/10/2012) |
| Slamet | 109.208E/ 7.242 S | 5 | Persistent degassing from the summit crater. | 3300–3400 | 206 | 66 | OMI | Carn et al. 2017 (2005–2015) |
| Merapi | 110.446E/ 7.540 S | 18 | Degassing from lava dome | 2800–2900 | 20 | 7 | DOAS scanner | This work (May-Jun.-Jul. 2016) |
| Arjuno Welirang | 112.575E/ 7.733 S | | Degassing associated with vigorous fumaroles—not measured | | ? | | | |
| Semeru | 112.922E/ 8.108 S | 16 | Continuous eruptive activity with intermittent strong events | 3500–3700 | 48 | 22 | UV-Camera | Smekens et al. 2015 (16–22/05/2013; 31/ 05–03/06/2013) |
| Bromo | 112.950E/ 7.942 S | 7 | Degassing via main crater | 2600–2700 | 166 | 2 | UV-Camera | Aiuppa et al. 2015 (20–21/09/2014) |
| Kawah Ijen | 114.242E/ 8.058 S | 4 | Degassing via a solfatara provoked by mining activity | 2300–2600 | 238 | 194 | UV-Camera | This work (12/05/2015) |

**Table 3 (continued)**

| Volcano Name | long/lat | Rank | Typical degassing status | Average plume height (m) | Mean SO₂ flux (Mg/d) | error (Mg/d) | Method/ Technique | Source/ (measurement date) |
|---|---|---|---|---|---|---|---|---|
| Rinjani | 116.470E/ 8.420 S | 12 | Degassing from intracaldera cone | 2300–2600 | 74 | 65 | OMI | Carn et al. 2017 (2005–2015) |
| Sangeang Api | 119.070E/ 8.200 S | 13 | Intermittent steam releases | 1700–1800 | 71 | 75 | OMI | Carn et al. 2017 (2005–2015) |
| | | | | Total SO₂ for Sunda arc | 1313 ± 539 Mg/d (~0.48 ± 0.20 Tg/yr) | | | |
| **Sangihe Arc** Soputan | 124.737E/ 1.112 N | 2 | Sustained degassing at the summit and frequent eruptions | 1800–2000 | 376 | 100 | UV-camera | This work (21/07/2014) |
| Lokon | 124.792E/ 1.358 N | 9 | Sustained degassing and frequent eruptions | 1200–1500 | 117 | 10 | UV-camera | This work (19/07/2014) |
| Karangetang | 125.407E/ 2.781 S | 8 | Degassing associated with lava dome | 1700–1900 | 120 | 55 | DOAS scanner | This work (24/07/ 2015) |
| Awu | 125.447E/ 3.689 N | 21 | Degassing via crater wall and from small intracrater lava dome | 900–1100 | 13 | 5 | DOAS scanner | This work (27/07/ 2015) |
| | | | | Total SO₂ for Sangihe arc | 626 ± 170 Mg/d (~0.23 ± 0.06 Tg/yr) | | | |
| **Banda Arc** Ebulobo | 121.191E/ 8.817 S | 26 | Degassing via summit fumarolic activity | 2100–2300 | 6 | 3 | UV-Camera | This work (02/10/ 2014) |
| Iya | 121.641E/ 8.891 S | 24 | Degassing via fumarole activity | 600–650 | 8 | 6 | DOAS Walking traverses | This work (01/10/2014) |
| Kelimutu | 121.820E/ 8.770 S | 30 | Negligible SO₂ emission through crater lake | 1550–1600 | 2.0 | 0.7 | DOAS walking traverse | This work (30/09/ 2014) |
| Rokatenda (Paluweh) | 121.708E/ 8.320 S | 14 | Minor gas release via lava dome | 700–1000 | 60 | 32 | OMI | Carn et al. 2017 (2005–2015) |
| Egon | 122.455E/ 8.676 S | 28 | Gas release via fumarolic activity at the summit | 1600–1700 | 3 | 2 | UV-Camera | This work (13/05/2013) |
| Lewotobi Lakilaki | 122.768E/ 8.537 S | 31 | Minor gas release from summit fumaroles | 1350–1400 | 2.0 | 0.7 | UV-Camera | This work (12/05/2013) |
| Lewotobi Perempuan | 122.781E/ 8.551 S | 20 | Degassing from a small intracrater dome | 1500–1600 | 15 | 10 | UV-Camera | This work (12/05/2013) |
| Ili Lewotolo | 123.508E/ 8.274 S | 11 | Gas release from the crater and fumaroles | 1400–1500 | 75 | 40 | UV-camera | This work (07/05/ 2013) |
| Ili Werung | 123.573E/ 8.532 S | 34 | Minor gas releases from fumaroles on the flank and crater wall | 550–580 | 1 | 0.8 | UV-Camera | This work (10/05/2013) |
| Batu Tara | 123.585E/ 7.791 S | 10 | Degassing from the main crater Eruptive activity: 2012–2015 | 750–800 | 102 | 51 | OMI | Carn et al. 2017 (2005–2015) |
| Sirung | 124.130E/ 8.508 S | 17 | Degassing via secondary craters. | 600–700 | 48 | 22 | DOAS scanner | This work (13/08/2015) |

**Table 3 (continued)**

| Volcano Name | long/lat | Rank | Typical degassing status | Average plume height (m) | Mean SO$_2$ flux (Mg/d) | error (Mg/d) | Method/ Technique | Source/ (measurement date) |
|---|---|---|---|---|---|---|---|---|
| Wurlali | 128.678E/ 7.125 S | 25 | Main crater hosts a crater lake Gas release from numerous solfatara zones | 700–800 | 8 | 6 | UV-Camera | This work (21/10/2019) |
| Serua | 130.017E/ 6.312 S | | Minor degassing from the summit | | ? | | | |
| Banda Api | 129.881E/ 4.523 S | | Minor degassing from the summit | | ? | | | |
| | | | | Total SO$_2$ for Banda arc | 330 ± 175 Mg/d (−0.12 ± 0.06 Tg/yr) | | | |
| **Halmahera Arc** | | | | | | | | |
| Gamalama | 127.330E/ 0.800 N | 19 | Degassing via a large fracture at the summit | 1650–1700 | 16 | 10 | DOAS scanner | This work (27/07/ 2014) |
| Gamkonora | 127.530E/ 1.380 N | 27 | Persistent minor degassing | 1350–1400 | 3.4 | 1.0 | DOAS scanner | This work (24/08/ 2018) |
| Ibu | 127.630E/ 1.488 N | 15 | Degassing associated with dome growth and explosions | 1200–1300 | 59 | 32 | DOAS scanner | This work (25/09/ 2018) |
| Dukono | 127.894E/ 1.693 N | 1 | Continuous eruptive activity, with variable intensity | 1300–1400 | 819 | 394 | DOAS scanner | This work (12/07/2015) |
| | | | | Total SO$_2$—for Halmahera arc | 897 ± 437 Mg (−0.33 ± 0.16 Tg/yr) | | | |

TOTAL FOR INDONESIA ARCHIPELAGO = 3166 ±1321 Mg/d (1.15 ± 0.48 Tg/yr)

Question mark (?) indicates persistently degassing volcanoes not yet measured.

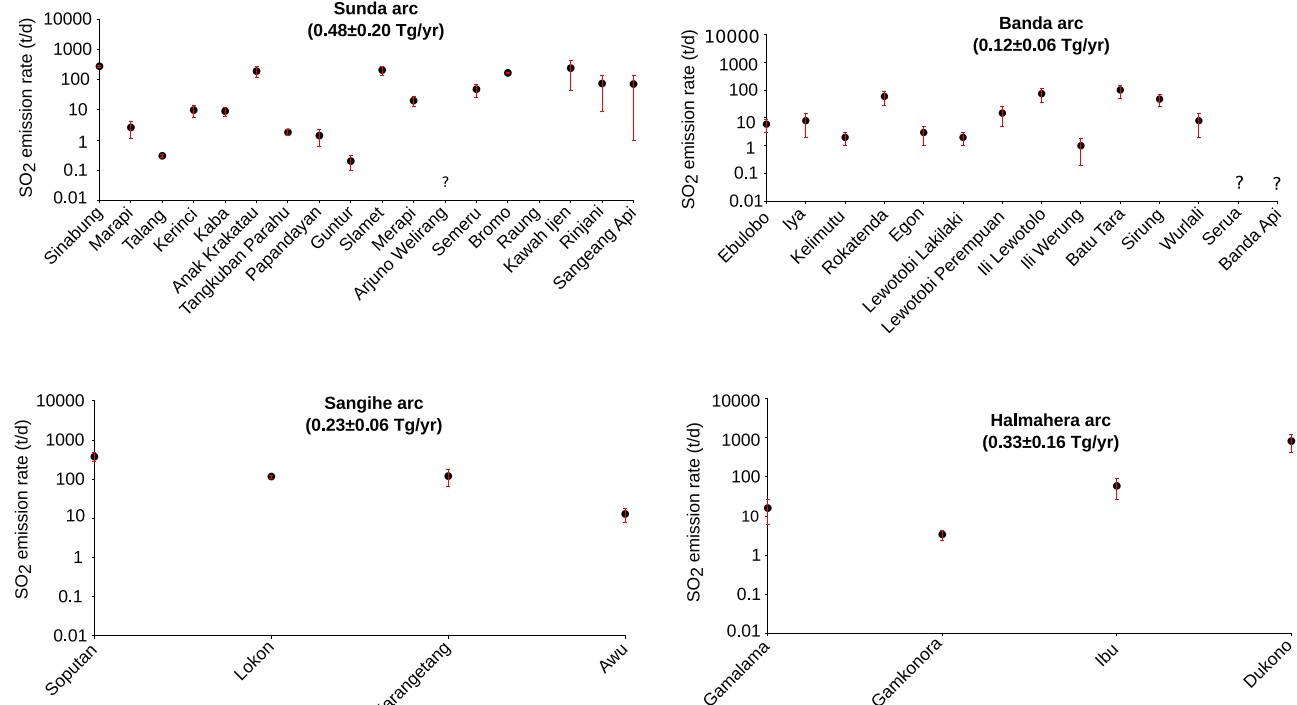

**Fig. 2 The main volcanic degassing points of Indonesia.** The $SO_2$ emission rates across the four volcanic arcs of Indonesia highlight the Sunda arc as the largest $SO_2$ contributor and Dukono is the strongest individual source. The question marks (?) denote the unmeasured sources and the error bars correspond to standard deviation.

**Table 4 Number of eruptive events per year and the corresponding $SO_2$ release per arc for 2010–2020 period for both from satellite and VEI results.**

| | 2010 | 2011 | 2012 | 2013 | 2014 | 2015 | 2016 | 2017 | 2018 | 2019 | 2020 | Total events/arc | Mean number of events/yr |
|---|---|---|---|---|---|---|---|---|---|---|---|---|---|
| Number of eruptive events | | | | | | | | | | | | | |
| Sunda | 5 | 4 | 2 | 6 | 9 | 6 | 3 | 8 | 8 | 6 | 3 | 60 | 5 |
| Banda | 0 | 0 | 4 | 3 | 2 | 2 | 0 | 1 | 0 | 0 | 1 | 13 | 1 |
| Sangihe | 1 | 3 | 3 | 1 | 1 | 3 | 2 | 1 | 2 | 1 | 1 | 19 | 2 |
| Halmahera | 1 | 2 | 2 | 2 | 2 | 2 | 2 | 1 | 2 | 1 | 1 | 18 | 2 |
| Total events/year | 7 | 9 | 11 | 12 | 14 | 13 | 7 | 11 | 12 | 8 | 6 | 110 | 10 |
| $SO_2$ emission per arc (Tg) | | | | | | | | | | | | Total $SO_2$ (Tg) | Mean Tg/yr |
| Sunda | 0.172 | 0.050 | 0.009 | 0.004 | 0.574 | 0.006 | 0.047 | 0.000 | 0.021 | 0.028 | 0.002 | 1.069 ± 0.184 | 0.097 ± 0.011 |
| | <u>0.641</u> | <u>0.044</u> | <u>0.022</u> | <u>0.082</u> | <u>0.824</u> | <u>0.235</u> | <u>0.056</u> | <u>0.151</u> | <u>0.821</u> | <u>0.722</u> | <u>0.118</u> | <u>3.715 ± 0.337</u> | <u>0.337 ± 0.021</u> |
| Banda | 0.000 | 0.000 | 0.001 | 0.005 | 0.000 | 0.000 | 0.000 | 0.000 | 0.000 | 0.000 | 0.062 | 0.068 ± 0.019 | 0.006 ± 0.003 |
| | <u>0.000</u> | <u>0.000</u> | <u>0.136</u> | <u>0.103</u> | <u>0.037</u> | <u>0.022</u> | <u>0.000</u> | <u>0.004</u> | <u>0.000</u> | <u>0.000</u> | <u>0.095</u> | <u>0.397 ± 0.05</u> | <u>0.036 ± 0.008</u> |
| Sangihe | 0.000 | 0.010 | 0.003 | 0.000 | 0.000 | 0.001 | 0.000 | 0.000 | 0.014 | 0.005 | 0.000 | 0.035 ± 0.005 | 0.003 ± 0.002 |
| | <u>0.095</u> | <u>0.133</u> | <u>0.133</u> | <u>0.018</u> | <u>0.004</u> | <u>0.118</u> | <u>0.099</u> | <u>0.004</u> | <u>0.114</u> | <u>0.017</u> | <u>0.019</u> | <u>0.755 ± 0.055</u> | <u>0.069 ± 0.018</u> |
| Halmahera | 0.061 | 0.060 | 0.063 | 0.047 | 0.099 | 0.080 | 0.203 | 0.104 | 0.055 | 0.248 | 0.407 | 0.143 ± 0.011 | 0.013 ± 0.003 |
| | <u>0.095</u> | <u>0.114</u> | <u>0.099</u> | <u>0.099</u> | <u>0.114</u> | <u>0.114</u> | <u>0.099</u> | <u>0.095</u> | <u>0.099</u> | <u>0.095</u> | <u>0.095</u> | <u>1.121 ± 0.008</u> | <u>0.102 ± 0.033</u> |
| Total Tg/yr | 0.233 | 0.121 | 0.075 | 0.057 | 0.673 | 0.088 | 0.250 | 0.104 | 0.090 | 0.281 | 0.473 | 1.314 ± 0.180 | 0.119 ± 0.045 |
| | <u>0.832</u> | <u>0.291</u> | <u>0.390</u> | <u>0.302</u> | <u>0.979</u> | <u>0.489</u> | <u>0.254</u> | <u>0.253</u> | <u>1.034</u> | <u>0.835</u> | <u>0.327</u> | <u>5.988 ± 0.310</u> | <u>0.544 ± 0.116</u> |

Underlined values are those obtained from VEIs.

wotobi Perempuan and Awu account for the remainder. We have also estimated explosive $SO_2$ emissions from Indonesia for the period 2010–2020 based on a simple scaling from reported VEI values and satellite records. The mean annual explosive-$SO_2$ obtained range between 0.12 and 0.54 Tg/yr, 63–81% of which is associated with the Sunda arc (0.10–0.34 Tg/yr), 5–7% (0.04–0.07 Tg/yr) the Banda arc, 3–13% (0.03–0.07 Tg/yr) the Sangihe arc and 11–18% (0.10–0.14 Tg/yr) the Halmahera arc. Combining the calculated passive and explosive $SO_2$ data suggests a total volcanic

$SO_2$ yield for the Indonesian archipelago of 1.27–1.69 Tg/yr. We consider this a representative figure, acknowledging that it is based on very limited temporal sampling of the volcanoes in question.

Our $SO_2$ inventory indicates a surprisingly modest $SO_2$ emission budget for Indonesian volcanoes, considering the 6000 km extent of the archipelago, four distinct volcanic arcs, 126 active volcanoes, and on the order of ten larger eruptions per year. For comparison, Ambrym (Vanuatu) and Kīlauea (Hawaii) volcanoes

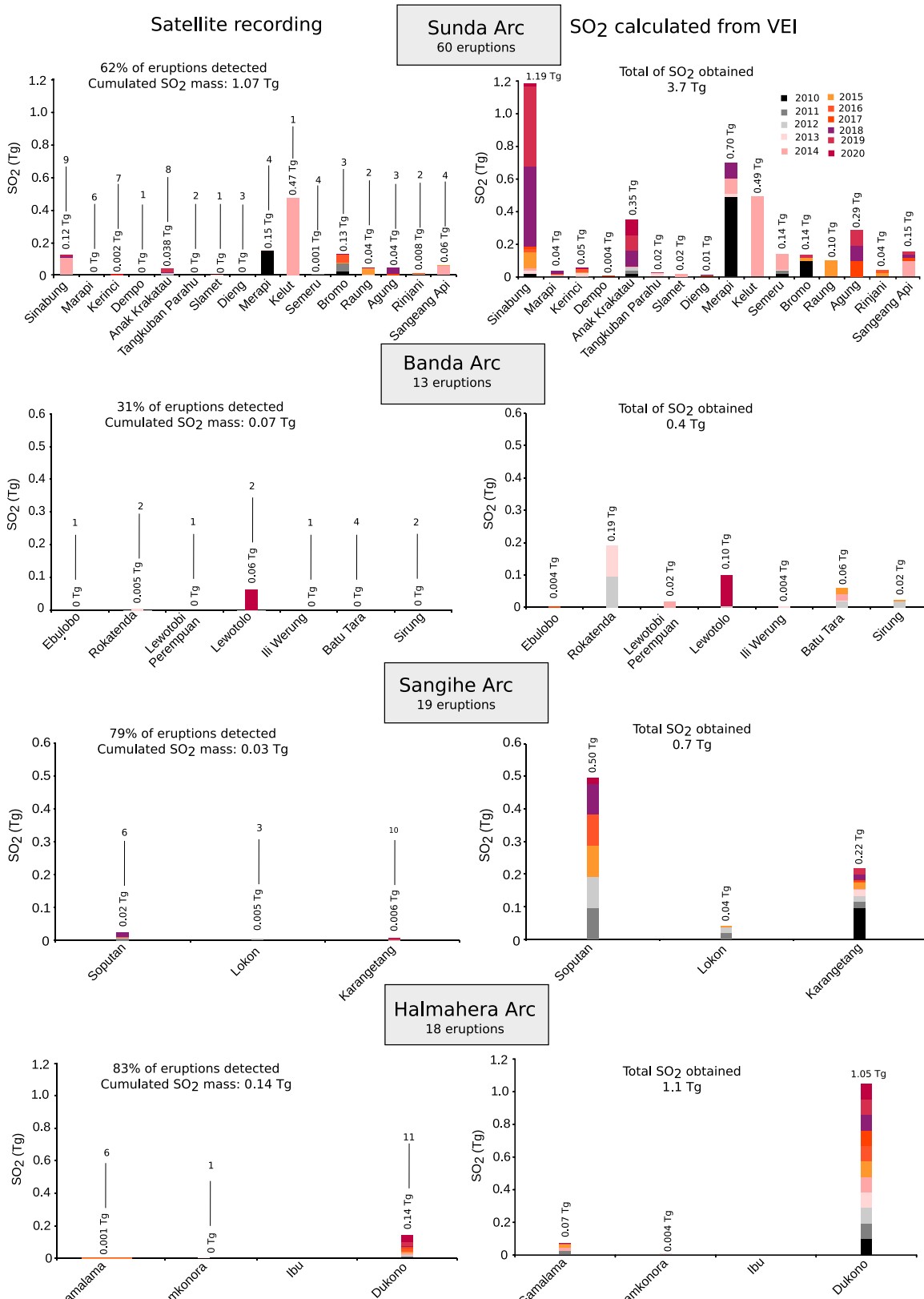

**Fig. 3 The explosive SO₂ released per volcano over the period of 2010–2020.** The names of the volcanoes that erupted over the decade are grouped by arc. The SO₂ mass per volcano obtained from satellite data are displayed on the left column whilst the right column shows the SO₂ amount obtained from the VEIs. The 0 Tg correspond to undetected eruptive emission by satellite sensors. The color code differentiates the years of observation and the height corresponds to the amount of SO₂ released per year. The number of eruptions per volcano is provided above each SO₂ mass value on the left column. Note that Dukono exhibits a continuous eruptive manifestation but only the largest event with ash fall on the nearby cities are considered.

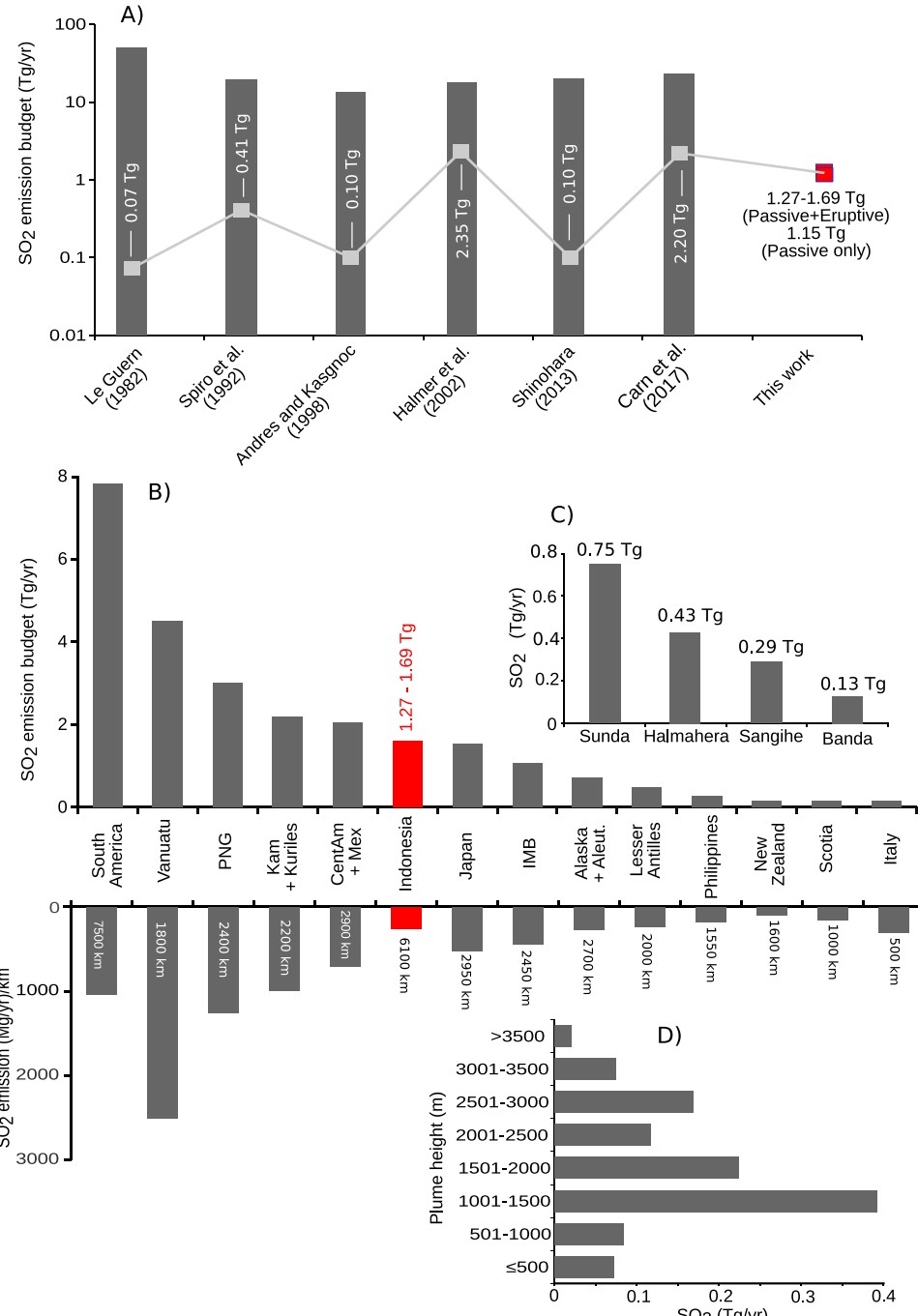

**Fig. 4 The new SO$_2$ flux results compared to other estimates. A** Estimates of the global volcanic SO$_2$ inventory that include contributions from Indonesian volcanoes, highlighted by gray square with the corresponding values. **B** The Indonesian SO$_2$ emission budget compared with other arcs (data from ref. [20]). The annual SO$_2$ emission per km of each arc[21] are shown for comparison. **C** The SO$_2$ emission budgets from the four Indonesian arcs. **D** Strength of passive SO$_2$ emissions by altitude (in 500-m bins) from the observations reported in Table 3.

alone have passively released more SO$_2$ into the atmosphere: 2.7 Tg/yr and 1.8 Tg/yr, respectively[3]. Several individual eruptions of the last 15 years also released comparable or higher SO$_2$ amounts compared with the annual Indonesian output, including Kasa-tochi (2.7 Tg) in 2008, Sarychev Peak (1.2 Tg) in 2009, Eyjafjal-lajökull (1.2 Tg) in 2010, and Nabro (4.5 Tg) in 2011[16]. This modest SO$_2$ emission budget also contrasts with the picture of renowned climate-changing Indonesian eruptions, including Agung 1963 (ref. [22]), Tambora 1815 (ref. [23]), Krakatau 1883 (ref. [24]), and Samalas 1257 (ref. [25]). More recently, the Galung-gung eruption of 1982–1983 (Java) yielded 2.5 Tg of SO$_2$ (ref. [26]).

While the overall SO$_2$ budget is unremarkable, the bulk of the emissions are into the free troposphere (Fig. 4D) likely to extend timescales of atmospheric processing and deposition of sulfur[27]. Previous studies have highlighted the contribution of sulfur deposition from volcanic plume to sulfur emissions from peat fires[28].

Factors controlling sulfur output are numerous and include deep source characteristics and chemical processes occurring during magma storage and transfer through the crust. Hydro-thermal scavenging and scrubbing of sulfur from magmatic-hydrothermal discharges is often invoked as a process for sulfur

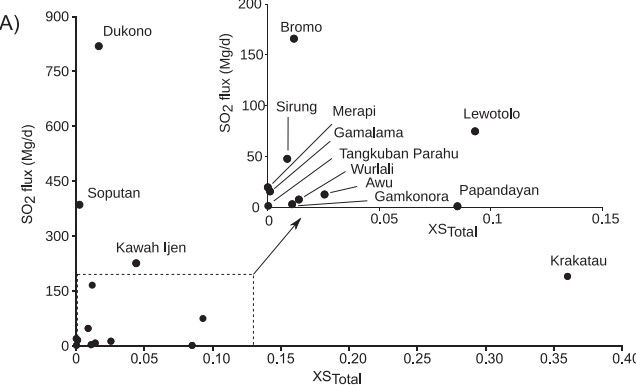

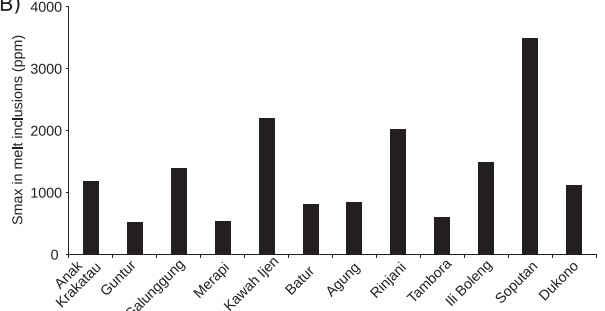

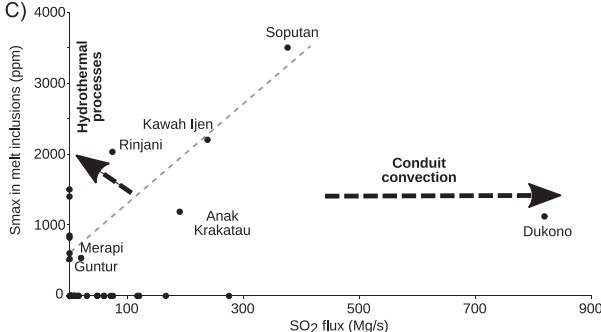

**Fig. 5 Sulfur content in fluids and melt inclusions versus SO₂ fluxes.**
**A** Variation of the sulfur content of volcanic fluids ($XS_{Total} = XSO_2 + XH_2S$) with the measured $SO_2$ flux along the Indonesian arc. References for fluid compositions are from Allard et al. 1981[56] (Krakatau); Poorter al. 1989[57] (Lowotelo); Giggenbach et al. 2001[58] (Merapi, Tangkuban Parahu, Papandayan); Clor et al. 2005[59] (Soputan); Aiuppa et al. 2015[60] (Bromo); Gunawan et al. 2017[61] (Kawah Ijen); Bani et al. 2017[62] (Sirung); Bani et al. 2017[15] (Dukono); Saing et al. 2020[63] (Gamkonora); Kunrat et al. 2020[64] (Gamalama); Bani et al. 2020[65] (Awu). **B** The record of sulfur in melt inclusions along the Indonesian arc in respect to the main degassing sources. **C** Relationships between the maximum sulfur content analysed in melt inclusions (MI) and the measured $SO_2$ flux. Note, we did not find melt inclusion values for data points with zero $SO_2$ flux and conversely. References for MI are: Vidal et al., 2016[25] (Rinjani); Mandeville et al., 1996[66] (Krakatau); Bani et al., 2017[15] (Dukono); Preece et al., 2014[67] (Merapi); de Hoog et al. 2001[68] (Guntur, Ili Boleng); Vigouroux et al. 2012[69] (Kawah Ijen, Galunggung, Tambora); Self and King, 1996[22] (Agung); Self et al. 2004[70] (Tambora), Kunrat, 2017[41] (Soputan).

depletion in volcanic fluids, significantly altering the magmatic signature[29]. Acidic crater lakes, which are numerous in Indonesia, are perhaps the most obvious manifestation of such processes[30]. Substantial sulfur deposits are known to be sequestered by volcanic lake systems and, conceivably, variations in climatic conditions, notably rainfall, across the archipelago could play a role in volcanic emissions to the atmosphere.

The subaerial sulfur output will depend on initial gas composition, the flow path, gas-wall rock heat transfer, and the effective water to rock ratio, all parameters that are difficult to constrain and which vary greatly between volcanoes. However, were hydrothermal scavenging and scrubbing leading mechanisms, one would expect flux strength to correlate with the concentration of sulfur in fluids. Instead, for the few volcanic centres for which, in addition to $SO_2$ flux, there are constraints on $H_2O$, $CO_2$, $H_2S + SO_2$ species in fumaroles, we find no correlation between the mass fraction of sulfur in the fluid, $XS_{tot}$ ($= XSO_2 + XH_2S$) and $SO_2$ flux (Fig. 5A), which presumably is a function of both the initial volatile content of the magma and degassing conditions. Dukono's gas, in particular, does not have higher sulfur than other Indonesian fumaroles (the case of Krakatau needs to be confirmed by more measurements). This indicates that $SO_2$ flux need not reflect particular enrichment/depletion in sulfur of the emitted gas, and implies the role of degassing vigour during the time interval considered (which can scale with conduit radius and presence of an open vent to the atmosphere). From this perspective, it is worth emphasising that to establish robust links between volcanic degassing and processes at depth, it requires comprehensive measurement of gas composition.

While hydrothermal sequestration of sulfur is likely to play a significant role in modulating subaerial emissions, we consider also whether arc scale differences in $SO_2$ emissions across the Indonesian archipelago might reflect geodynamic or source controls, as proposed for $CO_2$ in arc magmas worldwide[31]. The amount of $SO_2$ released per km of arc per year reveals the Halmahera arc as the strongest $SO_2$ source, followed by the Sangihe arc. The magmatic sources of these two arcs are sustained by the double subduction of the Molucca Sea plate that deepens to the west beneath the Sangihe arc, and to the east under the Halmahera arc[15]. Geochemistry of lavas sampled along these arcs indicates enriched magma sources in fluid-mobile elements and notable sediment contributions[32,33], which may play a role in subaerial sulfur budgets. The steepening of the subducted slab, the downward force from the Philippine Sea plate, and the westward motion of the continental fragments along the Sorong fault could have promoted fluid fluxes into the mantle wedge along the Halmahera arc[15]. To a first approximation, this peculiar geodynamic context may explain elevated $SO_2$ fluxes at both Halmahera and Sangihe arcs simply because of enhanced magmatic activity.

In contrast, the Banda arc stretches 2000 km but exhibits a remarkably low $SO_2$ emission, the weakest in our inventory. The arc is also characterized by anomalously low $^3He/^4He$ ratios[34] reflecting the arc collision with the Australian continental block and subduction of continental material that ultimately supplies less sulfur to the mantle wedge, compared to subduction of oceanic plate[35].

The Sunda arc is the largest $SO_2$ source, representing 43–48% of the total, however, its annual $SO_2$ emission per km of arc is modest compared with the Halmahera and Sangihe arcs, and with other arcs worldwide[20]. The magma source beneath the Sunda arc is sustained by subduction of the Indo-Australian plate. However, while deep sea drilling has revealed a 1400 m sediment column in front of Sumatra, 300 m in front of Java, and 500 m in front of Sumbawa[36], less than 15% of these sediment columns is subducted[37]. The mass transfer along the Sunda arc is dominated by an active frontal accretionary prism that strongly limits sediment subduction. Each year only $2.6 \times 10^7$ m³ of sediment is subducted beneath the Sunda arc, compared with the $1.8 \times 10^8$ m³/yr available, given the average subduction speed of 6.7 cm/yr. Furthermore, the sediment input from the Sunda arc is mostly trapped in the forearc basin and does not reach the trench. Given that subducted sediment can strongly contribute to the volcanic sulfur budget, it is possible that this active accretional

prism plays a key role in modulating the $SO_2$ emission budget of the Sunda arc by limiting the mass transfer of sediment-derived sulfur into the mantle wedge. However, as shown in the next section, such a variability in sediment contribution is not evident in variable sulfur abundance in magmas sustaining Indonesian arc volcanism.

Arc volcanoes are typically supplied by reservoirs in the shallow crust, which are in turn fed by basaltic melts rising from the mantle wedge and carrying an imprint of slab volatiles. Because they are hot, these mafic magmas have a higher sulfur carrying capacity than cooler silicic magmas, hence any volcano erupting mafic magmas should generally be associated with stronger sulfur emissions (though of course arc-scale variations in mafic melt sulfur content are possible). Detailed petrological studies of Indonesian volcanoes remain scarce, and only a few have had their sulfur content characterized via analysis of melt inclusions (MI). Figure 5B draws on these studies and shows the highest sulfur contents measured in MI along the Indonesian arcs. Drawing rough relationships from these data, basaltic MI have 2000–3500 ppm S, andesitic MI 800–500 ppm S, and rhyodacitic 200–300 ppm S, with no obvious geographical trends along arc being apparent, assuming that primary sulfur contents in the melt inclusions are little affected by post entrapment processes.

The most evident feature is that Dukono, the strongest $SO_2$ source we identify, has basaltic melt inclusions with rather low S content (1000 ppm) relative to other, currently weaker $SO_2$ sources, such as Rinjani or Kawah Ijen, which have MI with sulfur in excess of 2000 ppm. Similarly, Soputan emits half the $SO_2$ of Dukono, yet has basaltic MI with much higher sulfur content (3500 ppm). This is illustrated in Fig. 5C, which shows that a broad positive correlation between $SO_2$ flux and the maximum sulfur content, $S_{max}$, in MI. Only Dukono departs significantly from the trend indicated by other centres. This again argues for strong decoupling between the fertility of the immediate source of magma degassing (the crustal reservoir) and its ultimate surface manifestation. Such a behavior may simply reflect conduit dynamics, such as convection, which is strongly dependent on conduit radius[38–40], and which can sustain strong $SO_2$ degassing of an otherwise comparatively sulfur-poor reservoir. Alternatively, a more sulfur-rich magma may give rise to a low sulfur output simply because of a low magma influx that cannot sustain conduit convection. A critical parameter is the conduit radius, $R$, since magma (and gas) flux scales with $R^4$ (ref. [40]) such that small variations in conduit size can result in large fluctuation in sulfur flux.

The composition and temperature of the magma supplied to the conduit will also be important, owing to their influence on viscosity and rheology. Hotter and fluid material will promote not only higher rates of magma overturn in the conduit but also more efficient degassing of slowly diffusing species, such as sulfur, in silicate melts. From this perspective, systems lying above the dashed line toward Soputan, whose high $SO_2$ flux may be considered as directly related to its basaltic magma source[14] and high sulfur content in MI[41] (Fig. 5C), may reflect more pronounced scavenging by the aquifer/hydrothermal system overlying magma reservoirs, limiting sulfur emissions to the atmosphere. These local controls will be superimposed on any deeper source signatures and may even obliterate them, as exemplified here by Dukono.

The $SO_2$ emission budget of the Indonesian archipelago thus reflects a complex interplay between deep (geodynamic) factors that control primary magma compositions and their availability along the arc and superficial processes such as hydrothermal scavenging and conduit dynamics. Reservoirs regularly supplied by fresh hot magma may promote sustained and vigorous degassing via conduit convection, leading to strong $SO_2$ outputs even when sulfur complements in the melt are comparatively poor. In other words, the vigour of convection may largely compensate for, or even offset, any deep source deficiency in sulfur. The relatively low $SO_2$ output for the Indonesian archipelago documented here may appear in stark contrast with the record of explosive eruptions at several Indonesian volcanoes and their recognized global climate impacts (e.g. Tambora, Krakatau, Rinjani). These events, however, essentially reflect the long term accumulation of magmas and volatiles in closed crustal reservoirs, which cool and fractionate with little volatile loss, a process that differs from the persistent or passive degassing operating at open-conduit systems such as those we document here. An in depth knowledge of the petrology of volcanic products, and a robust characterization of emanating fluids, are both required if sound connections between the plumbing system and degassing are to be established at any active volcano.

## Methods

**Passive ultraviolet spectrometers.** We used two techniques, USB-controlled ultraviolet spectrometers and Differential Optical Absorption Spectroscopy (DOAS)[42] and ultraviolet cameras (UV-cam)[6]. The passive ultraviolet spectrometers were either carried beneath the plume on a moving platform[43] or located in a fixed position and attached to scanning optics[7]. Both approaches yield the $SO_2$ profile across the plume. The spectrometers used were the Ocean Optics USB2000 (280–400 nm, 0.5 nm FWHM resolution), USB4000 (290–440 nm, and 0.3 nm FWHM), and USB2000 + (290–440 nm and 0.5 nm). For traverse measurements, the spectrometer was connected via an optic fibre bundle to a vertically pointed telescope of 8 mrad FOV (Field Of View). The location of each recorded spectrum was obtained using a continuously recording GPS unit. The DOAS traverse setup requires no additional power supply since the spectrometer is powered by the laptop. We operated the equipment onboard a light aircraft, from a 4WD vehicle, and on foot.

For the scanning observations, we used a rotating window that accepts light from selected directions across the plume. The light that transits through the window is redirected to an embedded telescope by a 45° optical prism, then transmitted to the spectrometer via optical fibre. The rotating window was attached to a stepper motor controlled by the laptop via a microcontroller. The system was designed to perform a 180° scanning angle with a minimum step angle of 1.8°. The scanner required an external 12 V power supply. The scanning setup could be readily operated by one person. Spectra were acquired using Jscript executed by DOASIS software[44]. The script used in both traverse and scanning allowed optimization of the signal-to-noise ratio by automatically adjusting exposure time and numbers of co-added spectra[45]. This was particularly useful for scanning, given the change of light intensity with scan angle. Both traverse and stationary recording were carried out at distance varying between few tens of meters from the craters to around 5 km downwind, depending on the access difficulties, the plume size and the volcanic activity.

$SO_2$ column amounts (ppm m) were retrieved using standard DOAS calibration and analysis procedures outlined in ref. [43]. Reference spectra included in the non-linear fit were obtained by convolving high-resolution $SO_2$ and $O_3$ cross-sections with the instrument line shape. A Fraunhofer reference spectrum and ring spectrum, calculated in DOASIS, were also included in the fit. The optimum fitting windows were selected where they provided a near-random fit residual with minimum deviation. The total $SO_2$ column amount across the plume was then multiplied by the estimated plume speed to obtain the $SO_2$ flux. The plume velocities were measured mainly using videography and handheld anemometers, except in the case of airborne measurements where the plume speed was obtained by flying along and against the plume axis.

**Ultraviolet cameras.** The imaging setup consisted of two Apogee Alta U260 UV cameras. Each was coupled to a Pentax B2528-UV lens, with a focal length of 25 mm allowing a full angle FOV of around 24°. Immediately in front of each lens, a 10 nm (FWHM) bandpass filter was placed, one filter was centered at 310 nm (Asahi Spectra XBPA310) where $SO_2$ absorbs and the other at 330 nm (Asahi Spectra XBPA330) outside the $SO_2$ absorption region. Image acquisition and processing were achieved using Vulcamera[46]. For each pixel the optical depth (OD) was obtained according to the following equation:

$$OD = -\ln\{[(PA - DA)/(CA - DA)]/[PB - DB/(CB - DB)]\}$$ where A and B represent the camera with the 310 nm and 330 nm filters respectively, and P, D, and C represent plume, dark and clear images. To correlate the OD values with the $SO_2$ slant column densities (SCDs), four calibration cells with known amounts of $SO_2$ (94, 189, 475, 982 ppm.m) were used. Calibration images were acquired at the beginning of measurements and repeated with long series of measurements. The UV-cam was generally positioned with a view perpendicular to the plume transport direction, at distance between <1 km and 6 km depending on plume size. Plume

speeds were derived during data processing by following plume structures between two fixed lines perpendicular to plume transport direction[46].

**Uncertainties**. Uncertainties in DOAS and UV-cam $SO_2$ flux measurements are discussed in many past works, including the following ref. [47]. The dominant error in the retrieved $SO_2$ column amount is induced by the variability of light intensity and the distance between the plume and instruments. With increasing distance, light that has not traveled through the plume may contribute significantly to the signal. This leads to light dilution of the plume signal that can easily cause more than 50% underestimate in $SO_2$ emission rate[48]. To reduce this effect in the UV-cam measurement, we deployed the system during clear sky conditions, at distances <6 km, and performed calibration every hour during long series of measurements. UV-cam measurements were performed mainly in the late morning before the clouds started to formed, generally at 9–11 am.

For the DOAS measurement, we compensate for light intensity changes using an artificial constant dark, calculated from each recorded spectrum, in the 'UV blind' region (below 290 nm). Such corrections account for dark spectrum, offset and stray light. We estimate that the error in the column amount contributes ~0.01 to the squared variation coefficient of the total flux. We also assumed that the plume and transport direction is homogeneous and in a straight line since it is difficult to rigorously assess in this work. We, therefore, performed flux calculations for direction φ, φ±3, and φ±6. The mean contribution to the square variation of the total flux is in the order of a thousandth. These errors are negligible in comparison to uncertainties in the plume speed that resulted from the complexity of wind field around volcanoes and frequent variations in both time and space. The plume transport speed relative error is conservatively assumed to be about 30–35%, which is towards the higher end of the range of past estimates[49]. These errors are applied to each traverse and profile then the mean value is calculated for each series of measurement with the corresponding standard deviation. The global estimates for the arc is the sum of the mean values.

## Data availability

The data that support the plots within this article are available from the corresponding author upon reasonable request.

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

## Acknowledgements
This work was carried out under the collaboration between IRD (Institut de Recherche pour le Développement) and CVGHM (Center for Volcanology and Geological Hazard Mitigation). Field expeditions were supported by JEAI-COMMISSION and ANR-DOMERAPI projects. B.S. activity was supported by LabEx VOLTAIRE (LABX-100-01). This is Laboratory of Excellence Clervolc contribution n° 542.

## Author contributions
The study was designed by P.B. and C.O. Data processing protocols and instrument calibrations were supervised by V.T. Field measurements were carried out by P.B., S.P., U.B.S., H.A. and M.M. Data analyses were performed by P.B. The paper was primarily written by P.B., C.O. and B.S. with input from all authors.

## Competing interests
The authors have no competing interests.

## Additional information

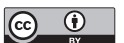

