## [Peer Review File · Nature Communications]

Modest volcanic SO₂ emissions from the Indonesian archipelagoEditorial Note: In their review of the first version of this manuscript, Reviewer #3 added some comments to the manuscript file. These comments, excluding minor textual revisions, have been copied into this Peer Review File on page 8.

REVIEWER COMMENTS

Reviewer #1 (Remarks to the Author):

Summary

The manuscript presented here details the results of recent measurements of the emission rates of sulfur dioxide (SO₂) from 47 volcanoes from the Indonesian archipelago, along with a detailed compilation of the existing complementary datasets in the literature. This is indeed a very valuable contribution to the literature and will be of great interest to the volcanological community. Passive degassing is poorly documented, especially in Indonesia, and this effort helps constrain the magnitude of passive degassing and its importance in global volcanic degassing budgets. The study finds a surprisingly modest level of degassing for the archipelago. The manuscript is well written and the conclusions are supported by the evidence presented. I only have a few minor comments and questions, detailed below, where I believe the authors could provide a more extensive discussion of some of the uncertainties within their methods, or additional evidence to support some of the suggested reasons for the low levels of observed degassing.

General comments

Extrapolation of sporadic measurements: Most of the estimates for passive degassing are based on measurements performed over very short periods. I would welcome a discussion of the uncertainties in how representative such measurements are likely to be at any given volcano. Some volcanoes exhibit cyclic behavior in their emissions for example (Semeru, Sinabung, Merapi) on timescales from hours to weeks. Have these variations been accounted for? If so how? And how would it impact the reported average emission rate?

Estimates of explosive contributions: The use of the formula relating VEI to SO₂ emissions from (Carn et al. 2016) presents a few difficulties, which I feel have not been addressed in the discussion, and the associated uncertainties have not been stated:

1. The formula was derived from satellite observations. Such observations are unlikely to have included many of the VEI <3 eruptions so common in Indonesia. This is because of several reasons: 1) the small magnitude of the eruptions, 2) the plume altitude often within the troposphere, and 3) the location of the eruptive centers close to the equator, all of which are factors making the satellite retrieval method more difficult. The authors are using a formula which was derived from data specifically excluding the type of eruptions they are trying to assess.
2. There is considerable spread in the data used to derive the formula. The classification of each event into a discrete VEI (integer class, not continuous), and the large range of observed emission rates within each VEI category introduces large uncertainties when using the formula. The reported errors in Table 5 seem to be related to the statistical averaging of multiple years, and do not include what is likely a +/- 40% error from simply using the formula.

Although the focus of the paper is clearly on passive emissions, it may be helpful to include a few direct comparisons between documented satellite retrievals over the period 2012-2019, and the approximations taken from the formula. This would provide confidence that the formula is a reasonable approximation, and that the explosive contributions can indeed be estimated in this way.

The influence of conduit radius: One of the main messages from the manuscript as I understand it is that the vigor of convective processes within the conduit (largely controlled by conduit radius) is likely the most important factor in explaining which volcanoes exhibit the largest emissions of SO₂. Is there any corroborating evidence suggesting that convection is indeed more vigorous (or that conduit radius is larger) at volcanoes where larger SO₂ emissions are detected? For example, heightened volcanic tremor or inflation/deflation cycles?

Specific comments

Table 1 is a nice addition, and a helpful compilation of the literature on the subject. But is a bit busy at the moment, and it may be beneficial to streamline it. I would suggest adding designated columns for: 1) the publication date of the study; 2) the time period covered for the yearly estimates; and 3) a succinct descriptor of the methods (i.e. COSPEC vs DOAS vs Petrological estimates, etc.). The reader could use this table much more efficiently when reading the manuscript.

L77-78: "... which constitute the main volcanic degassing sources in Indonesia (Table 2, Fig.1)" Consider adding a citation or justification for this statement. Measuring degassing rates from fumarolic fields is notoriously difficult given the lack of a point source. Recent efforts with UV cameras have started to place constraints on such emissions (e.g.; Stebel et al. 2014). This kind of estimates probably suffers the most drastically from the lack of measurements mentioned by the authors for all Indonesian volcanoes. I understand the choice not to include them in this study for that reason. If there is evidence to support their relative contribution is indeed negligible, please include it here.

Figures 1-3 could be combined into one figure. Perhaps even leaving three panels within it for clarity. But most of the information contained in each of these figures is redundant to some extent.

Table 3: When a volcano is labeled as "Quiescent, no gas" or as having "Negligible degassing", no measurement method is associated. Have any (obviously unsuccessful) measurements been performed to determine the lack of emissions? If so, it would be helpful to mention it. If not, what is this determination based on (gas sensors, lack of reports of sulfur smell by nearby communities)?

Figure 4: uses tonnes/day as opposed to Tg/yr. Is there a reason to use one unit over the other? I understand some measurements campaigns only consisted of one day, but it would be clearer for the reader if all values were reported as the extrapolated averaged values with a consistent unit throughout the manuscript (either t/d of Tg/yr).

References:

Carn SA, Clarisse L, Prata AJ (2016) Multi-decadal satellite measurements of global volcanic degassing. *Journal of Volcanology and Geothermal Research* 311: 99–134. <https://doi.org/10.1016/j.jvolgeores.2016.01.002>

Stebel K, Amigo A, Thomas H, Prata AJ (2014) First estimates of fumarolic SO₂ fluxes from Putana volcano, Chile, using an ultraviolet imaging camera. *Journal of Volcanology and Geothermal Research* 300: 112–120. <https://doi.org/10.1016/j.jvolgeores.2014.12.021>

Reviewer #2 (Remarks to the Author):

This paper presents an updated volcanic sulfur dioxide (SO₂) emissions inventory for Indonesia, based on a combination of new ground-based ultraviolet (UV) remote sensing measurements and some prior satellite and literature data. With a high density of active volcanoes and a history of producing large, sulfur-rich (climate forcing) eruptions, Indonesia is a region of broad interest to the volcanological and atmospheric science communities. A highlight of the study is the collection of new ground-based SO₂ flux data for 47 Indonesian volcanoes; a significant effort in the sprawling Indonesian archipelago which undoubtedly involved some challenging fieldwork. Based on their new SO₂ emissions inventory, the authors conclude that Indonesian volcanic emissions are modest relative to many other active volcanic arcs, and propose various petrological, geodynamic and shallow volcanic processes that could reconcile this observation with Indonesia's history of sulfur-rich explosive volcanism. Their overall conclusion is that the latter is likely due to long-term gas accumulation in upper crustal magma reservoirs.

This is a paper of broad interest with potentially significant implications for the identification of possible future sites of climate-forcing eruptions, and the application of an arc-scale volcanic SO₂ inventory to gain insight into the volcanic processes affecting volatile fluxes is novel (noting that Indonesia comprises multiple arcs). It might be among the most comprehensive volcanic SO₂ emissions inventories for any arc or volcanic region (especially one as extensive as Indonesia). However, I have some concerns with the SO₂ emissions inventory in relation to previous measurements (see below), and I'm not sure that the inventory alone merits publication in Nature Communications (though it is a significant body of work). The significance hinges on whether the Indonesian emissions are indeed modest relative to other arcs (and here the quality and extent of the SO₂ data from other arcs is also important) and if it is appropriate to draw conclusions about long-term degassing or gas accumulation from such a short temporal sample. The abstract mentions Ambrym (one of the strongest global volcanic SO₂ sources of recent years; ~2 Tg/yr SO₂) but Ambrym's SO₂ flux recently declined to negligible levels after a 2018 eruption, demonstrating that volcanic SO₂ emissions can change quite dramatically on short timescales (Kilauea is a similar case). Overall, I feel that the conclusions are quite speculative, but with some revisions and attention to the concerns below it could be acceptable for Nature Communications.

1) One of the main issues relates to the magnitude of the newly reported SO₂ emissions in comparison to previous satellite measurements, and whether the Indonesian volcanic SO₂ emissions are in fact as 'modest' as reported here. Several of the 47 volcanoes with new SO₂ flux data have SO₂ emissions previously estimated from satellite measurements, and there are some notable discrepancies which could be discussed in more detail. For example, the reported SO₂ fluxes for Kerinci, Ebulobo, Lewotolo, Sirung and Slamet are significantly lower (an order of magnitude in some cases) than space-based estimates. The paper reports SO₂ emissions for Slamet measured in 1991 (~30 tons/day), whereas there are more recent satellite data indicating higher SO₂ fluxes (average of ~200 tons/day). Where both satellite and ground-based data are available at a volcano, the approach here is to use the latter, and hence the implication is that the satellite measurements overestimate SO₂ fluxes, whereas the converse is generally regarded to be true (i.e., satellites underestimate SO₂ fluxes due to limited sensitivity). The authors do point out that the higher satellite-based SO₂ fluxes may be due to the inclusion of eruptive emissions, but this seems irrelevant if the goal is to quantify total volcanic SO₂ output. It would be nice to see more effort to reconcile the ground-based and satellite measurements. This is also relevant to the comparison with other arcs (Figure 6), since the SO₂ data for all the other arcs are also derived predominantly from satellite measurements. If the same reasoning is applied to other arcs (i.e., satellite measurements overestimate the fluxes) then perhaps SO₂ emissions from other arcs are also lower than shown in Fig. 6? This might have implications for the overall conclusions of the study. The analysis is also quite sensitive to the arc lengths, which should be provided.

2) A related issue is the timescale of the measurements. This is a common (and often unavoidable) issue with ground-based volcanic gas data which are typically collected during brief measurement campaigns. Many of the volcanic SO₂ emissions reported here represent a single day (or a few hours) of measurements, and it raises the question of how representative these are of long-term degassing rates, given that one of the overall conclusions of the paper relates to long-term gas accumulation. On the other hand, some of the satellite measurements span more than a decade (since 2005) and hence arguably provide greater insight into temporal variations in degassing, albeit still on a relatively short timescale. I wondered if there were any other data (e.g., seismic data, heat flux) available from the volcanoes studied here that might be useful for analyzing temporal variations in activity.

There is certainly scope to extend the study beyond ~2010-2020. Satellite measurements are available since 2005 for passive and eruptive volcanic SO₂ emissions and since 1978 for eruptive emissions. The latter includes significant eruptions of Indonesian volcanoes such as Galunggung, Colo, Makian [Kie Besi] and Banda Api in the 1980s-90s (notably, these volcanoes currently appear to have negligible SO₂ emissions). Some of these data might further support the paper's conclusions about gas accumulation.

3) Another question is how scrubbing of SO₂ and emission of other sulfur species (especially H₂S) might affect the Indonesian volcanic sulfur budget. Most Indonesian volcanoes are 'wet' and as the authors point out there are numerous acidic crater lakes. The authors discuss scrubbing and the total sulfur content of some Indonesian volcanic gases on page 22, but the compositional data

(Fig. 7) are limited and do not include many of the major volcanic sulfur sources. If possible, I think an effort to constrain volcanic H₂S emissions in Indonesia would be valuable. The authors acknowledge the importance of scrubbing or hydrothermal sequestration (e.g., L314) but seem to reject it as a major influence on arc-scale SO₂ emissions. Variable degrees of scrubbing of volcanic SO₂ in 'wet' vs 'dry' arcs (e.g., much of South America in the latter case) could perhaps partly explain the data shown in Figure 6.

4) The petrological data that underpin some of the main conclusions (Fig. 8, 9) are somewhat limited (as the authors acknowledge) and melt inclusions (MI) are not necessarily a robust indication of primary melt sulfur contents, as they can be affected by various processes (pre-entrapment degassing into a magmatic volatile phase, post-entrapment modification etc.). These issues should at least be acknowledged. Furthermore, there is no discussion of how the other more abundant volatile species (H₂O, CO₂) could play a role in the volcanic processes – the emphasis here is on SO₂ but the other volatiles play a much more significant role in driving volcanic activity.

Further comments:

L35: I think 'arguably' can be deleted here.

L35-47: Perhaps a little odd to have no citations in the first two paragraphs. There are good review articles that could be cited for all the statements given here.

L46: in terms of satellite measurements, temporal and spatial gaps are much less of an issue now with high-resolution sensors such as Sentinel-5P/TROPOMI. The more significant current challenges are the low sensitivity to weaker volcanic SO₂ sources and the challenge of processing the large volume of data.

L48: the second sentence of this paragraph seems out of place here?

L65: Table 1 could perhaps be condensed as I'm not sure all the information is needed – perhaps the more relevant data could be shown here (i.e., the inventories that specifically report data for Indonesia), with a larger compilation in supplementary material? There have certainly been other reviews of the many volcanic SO₂ inventories compiled over the years. Also note that the Carn et al. data cited (last line of Table) spans 2005-2015 (not 2014), but the NASA satellite-based inventory is also updated annually so there are now additional SO₂ emissions data for 2005-2020. Based on recent activity, a couple of new Indonesian SO₂ sources have been added to the database (Arjuno-Welirang and Agung), although this has minor impact on the total Indonesian SO₂ emissions (average ~2.3 Tg/yr).

L73-75: although the focus is naturally on the Type-A volcanoes, I do wonder about possible H₂S emissions from the numerous Type-B/C volcanoes. Could this be a significant source of sulfur degassing in Indonesia?

L77: 'Halmahera' (typo). Plus, should it be Sulawesi-Sangihe for consistency with Table 2?

L92: 'subaerial'

L97: Batu Tara is also listed in the satellite-based inventory, though the SO₂ emissions are merged with Lewotolo. Arjuno-Welirang has recently been added to the database too.

L139: Recommend providing the alternate name for Rokatenda (Paluweh) to avoid any confusion.

L144-146: Based on satellite observations, evidence suggests that SO₂ emissions from Batu Tara must be more significant than Banda Api. There is a substantial SO₂ signal over Batu Tara in some years, but no passive degassing has been detected from space at Banda Api to date.

L158: Kie Besi more commonly known as Makian?

L180: although there is some agreement, these rankings would look somewhat different if the satellite SO₂ inventory for Indonesia (2005-2015 or 2005-2019) was also considered. Summing the average SO₂ fluxes in 2005-2019 for the 19 Indonesian volcanoes with emissions detected from space, the total SO₂ flux is ~6200 Mg/d (~2.3 Tg/yr), i.e., more than twice that reported here.

L207: should it be ref. 3 cited here, rather than ref. 15?

L210: it may be partly true that the satellite measurements include more vigorous eruptions, though many of the larger eruptions are filtered out by using a threshold SO₂ column amount in the satellite data analysis. Furthermore, eruption clouds tend to drift away from the volcanic source, whereas the passive emissions are derived only from satellite data collected close to the volcano in question. But regardless of the origin of the SO₂, both passive and eruptive emissions contribute to the overall arc flux, hence I don't see the inclusion of eruptive emissions as an issue here.

L213-214: without seeing a detailed list of the eruptive events, it is difficult to assess the accuracy of this statement, unless the satellite data were analyzed as part of this work.

L225: Note that some of the explosive eruptions (e.g., multiple Sopotan and Sinabung eruptions) were detected in the satellite data and would be listed in the NASA database of eruptive volcanic SO₂ emissions (https://disc.gsfc.nasa.gov/datasets/MSVOLSO2L4_4/summary). Hence, I don't think it is necessary to use the VEI-SO₂ relationship (which has significant uncertainty) for all the explosive eruptions, if actual measurements are available.

L259: since arc lengths are a factor in the analysis, the arc lengths used in the paper should be provided. The results in Fig. 6 are quite sensitive to this parameter and more details on how it was measured or obtained (including any uncertainty) would be useful.

L278-279: though ironically, both Ambrym and Kilauea have had much reduced SO₂ emissions since 2018-19, which shows the dynamic nature of volcanic degassing and the importance of the timescale of observations.

L280-282: the 1982-83 Galunggung (Java) eruption produced ~2 Tg SO₂, so there have also been relatively recent Indonesian eruptions with comparable output to those listed here.

L355-356: presumably Sopotan's frequent explosive eruptions are linked to its volatile-rich magma (not just sulfur but also H₂O, CO₂).

L357: I think it is clear that Dukono is a special case among the currently active Indonesian volcanoes, since its activity (with frequent emission of volcanic ash; presumably juvenile magma) seems to shift between continuous eruption and passive degassing. Perhaps during continuous eruptive activity there is less conduit convection. Given that eruptions usually involve a greater flux of magma, this would certainly explain the high SO₂ flux at Dukono.

L405: although the spectroscopic techniques used to measure the volcanic SO₂ fluxes are described here in general terms, it is a bit difficult to assess the data quality at individual volcanoes without further information on measurement conditions (e.g., were the plumes optically thick/condensed, possible aerosol impacts., etc.)

Figure 4-5, 8: these figures could be improved. Bar charts may not be the best way of presenting these data, as there is a lot of white space and the text is small.

Figure 6: A) a log scale might work better here as the Indonesian volcanic contribution is difficult to see. And perhaps also show only those inventories which report data for Indonesian volcanoes?

Figure 9: Are the points plotted along the x-axis (i.e., ~zero or very low S in MI) actual measurements?

Simon Carn
Michigan Tech

Reviewer #3 (Remarks to the Author):

The authors present a comprehensive compilation of SO₂ flux measurements from the Indonesian volcanic arc and highlight the important observation that, despite abundant volcanism and a record of climate-forcing eruptions, the arc-scale SO₂ flux is actually relatively modest in a global context. These data then promote an insightful discussion on the various controls on sulfur emissions, and how outgassing fluxes may be decoupled from the initial sulfur content of primary melts.

The research is rigorous, represents a substantial field effort on behalf of the authors, and will be of broad interest across the community.

I have made my comments directly on the manuscript file, so please see attached. However, I outline a few general questions here:

- Uncertainties: In data-rich papers, such as this one, the propagation of uncertainties is really important. Throughout the paper, I would encourage the authors to explain clearly what each reported uncertainty represents (i.e. a measure of variance on repeat measurements, or an

absolute uncertainty associated with the measurements themselves, such as wind speed), and how these are carried through to the final arc-scale fluxes. These error bounds should be displayed on figures, even when they are in the form of bar charts. On this note, I might suggest that some of the data might be better represented with scatter plots rather than bars, when illustrating discrete values.

- Figures and tables: There are some repetitions between sequential figures and tables; for example, between figures 1-3 and between tables 3 and 4. In the interests of being concise, but also grouping similar datasets together, I would encourage the authors to think about whether any of these could be combined, either by merging or by creating multi-part (a and b) figures.

- The phrasing "sulfur budget" or "degassing budget" is used frequently throughout the paper. To me, a budget requires an evaluation of inputs and outputs... the authors are presenting only outputs, and therefore fluxes. Have a think about whether budget is really the right term here.

- The units reported switch between Tg and Mg, and once kt, throughout the paper. Consider keeping the units consistent for each of comparison.

- Comparing time-averaged vs "instantaneous" measurements. Carn et al 2017 report time-averaged fluxes over a decadal period, whereas the measurements in this study are (I think) predominantly campaign-based? It is an interesting question to what extent long-term averages can be compared to "instantaneous" flux measurements, and this could have been brought into the discussion more strongly as I think it is very relevant to the points being made.

- In the discussion, statements are made regarding the link between sediment inputs into the subduction zone and the sulfur content of the magmas that yield the emitted gases. But is the relationship between sediment flux, primary melt S contents, and melt inclusion S contents this simple though? The S in melt inclusions is affected by the sulfide saturation of mantle source, sulfide saturation and sequestration during magma ascent and storage, and importantly fluid exsolution prior to melt inclusion entrapment... can S loss prior to MI entrapment be discounted as a contributing factor to the lower than expected MI concentrations for Dukono, for example? Are there any constraints on entrapment pressures in the original studies where these data are sourced from?

- In the discussion, it would be good to see a slightly more nuanced discussion regarding the interplay between temperature, composition, and redox on sulfide saturation, and thus dissolved sulfur contents in magmas of different compositions.

- When discussing figure 9, are there other processes, such as gas fluxing of a segregated volatile phase, that could explain high SO₂ fluxes without the need to invoke extensive convection, particularly for more evolved compositions? Where do you envisage this degassed magma accumulates if not erupted?

Many thanks for the opportunity to review this manuscript, I particularly support the authors' emphasis on the need to integrate petrologic observations with surface gas emission measurements; this will be really important moving forwards.

All the best
Emma Liu

Reviewer #3 (Remarks to the Author, from manuscript file):

Line 46

There is not a single citation in the opening two paragraphs. Please consider acknowledging some of the key studies that have led these developments.

Line 49

Reference here to those compilations?

Line 53-54

Try to be consistent whether you indicate Tg SO₂/yr or Tg/yr

Line 65

You might consider defining the acronyms in the table caption for those unfamiliar with OMI, TOMS, COSPEC etc.

Line 85

There is some repetition between figure 1 and 2. Consider if they might be better combined into either a single or a multi-part (a and b) figure?

Line 92

Conducted measurements at?

Table 3

Consider saying in the figure caption explicitly what this error represents (e.g. the standard deviation of repeat measurements? The absolute measurement uncertainty?)

Line 119

It is difficult to compare between the different plots as the y-axes are all scaled differently. Could you use the same scale for a, b, and c... and then for d perhaps note the difference in scale in the caption?

Please indicate the error bounds on these bars, and in the titles

Line 125

Uncertainties?

Line 140-141

This phrasing is a little clunky; it's either negligible or it's not...

Line 144

Degassing fluxes?

Line 152

What defines a 'notable' contribution?

Line 154

This value alone is not really a budget, it a flux

Line 159

Are all measurements for passive emissions, or are some eruptive (explosive?)?

Line 165

Reporting all the SO₂ fluxes in this paragraph for each volcano mentioned, repeats the information described in the previous section.

Could you instead give the %?

Line 167

Could you instead give the %?

Line 179

There is a lot of repetition between table 3 and 4. Consider how the information in these two tables could be combined.

Line 182

Either demonstrate the differences to be statistically significant, or use a different word here (e.g. substantial)

Variations in total SO₂ output?

Line 185

What do you mean by "low source strength"?

Line 187

Less sulfur than what...? (compared to subduction of oceanic plate?)

Line 190

You might want to consider explaining "double subduction"

Line 189-197

The text highlighted in green should be saved for the discussion, where you can expand on each point a bit more. Indeed, some of this text is repeated in the discussion already.

Line 207

Can you indicate in the figures which of your measurements correspond to passive vs eruptive (explosive?) degassing?

Line 208

Such as? Please provide some examples of where the two datasets diverge

Line 209

Carn et al report time-averaged fluxes over a decadal period, whereas the measurements in this study are (I think) campaign-based? It is an interesting question to what extent long-term averages can be compared to "instantaneous" flux measurements.

Line 227

How was this uncertainty determined from the equation?

It's slightly confusing that all earlier sections reported fluxes in Mg, whilst here it is Tg. You might just want to make that distinction clearer, or use consistent units throughout.

Table 5

kt or Tg?

Line 240

Please indicate the error bounds on these bars, and in the titles

Line 248

It is again a little confusing to be combining Tg and Mg on the same figure. Consider keeping a consistent unit?

Line 250

Total flux?

Line 251

Upper bound?

Line 253

Does this uncertainty include the propagated uncertainty on the mean explosive flux of 0.54?

Line 255

"source strength" does not quite make sense to me in this context.
Total emissions from...?

Line 262

It is either 20 or it's not. Give the actual number of sampled sites?

Line 269

These values don't all match the % given in the text on page 18.

Line 273

Consider your use of "budget" throughout. To me, a budget requires an evaluation of inputs and outputs... you are presenting only outputs, and therefore fluxes

Line 275

Give locations

Line 276

Uncertainties on these fluxes and mass loadings?

Line 283

Please expand on these points

Line 288

I can see the point you are trying to make here, but am reading between the lines. Can you make the link between sulfur sequestration in volcanic lakes and the influence of rainfall on sulfur emissions more explicit?

Line 290

Are you still referring specifically to the sulfur flux through volcanic lakes, or more generally?

Line 292

What exactly are you referring to by "this", as you have outlined several processes in the previous sentence

Line 293

Which presumably is a function of both the initial volatile content of the magma, plus the conditions of degassing (depth/pressure in particular)?

Line 295

How about the relationship between X_{SO_2} and SO_2 flux?

Line 298

What do you mean by degassing vigour?

Line 301

Such as?

Also, surely volcanic degassing is itself a process at depth? (this is why I tend to reserve degassing for the process of exsolution, and outgassing for the release of gas to the atmosphere)

Line 310

In modulating subaerial emissions?

Line 314

Explain what you mean by double subduction?

Line 315

Can you expand on this discussion, to link these observations explicitly to how they would influence subaerial sulfur emissions?

Line 319

From where to where?

Line 321

Can you be more specific about what you mean?

e.g. Greater degrees of melting in mantle source region? Higher magma fluxes to shallow crust?

Line 324

Can you expand on this to explain how this is evidenced by the He isotope composition?

Line 334

All of it? Or most?

Line 335

In what way? It would add greatly to the clarity of your discussion if you added further specific detail about the mechanisms for each of the controls you discuss

Line 337

... by reducing the mass transfer of sediment-derived sulfur into mantle wedge source regions, and thus limiting initial melt S contents?

Where is "below"? Perhaps refer to a specific figure or section instead?

Line 338

Is the relationship between sediment flux and melt inclusion sulfur contents, for example, this simple though? The S in melt inclusions is affected by the sulfide saturation of mantle source, sulfide saturation and sequestration during magma ascent and storage, and potential deep fluid exsolution...

Line 341

It would be good to see a slightly more nuanced discussion here, regarding the interplay between temperature, composition, and redox on sulfide saturation, and thus dissolved sulfur contents in magmas of different compositions.

Line 346

It would be helpful to indicate on Figure 8 which volcanoes are associated with each composition. (perhaps different coloured bars?)

Line 349

But can you discount volatile loss prior to MI entrapment?

Line 352

I presume that the best fit line shown on Figure 9 does not include those volcanoes where the max S in MI is 0? If they were included, you might argue that the trend is not so apparent.

Line 354

Is this interpretation based only on the datapoint for Dukono?

Line 356

What about for the more evolved magmas, as their viscosities would limit convection? And where do you envisage this degassed magma accumulates if not erupted?

What about gas fluxing by a segregated volatile phase?

Line 360

Do you observe a relationship between SO₂ flux and conduit radius? Is enough information on conduit radii available to test this?

Line 363

Rheology would also encompass viscosity

Line 364

Are there other processes, such as gas fluxing, that could explain the high SO₂ fluxes without the need to invoke extensive convection?

Line 365

What does the dashed line represent? Is it a linear regression line? Or simply to guide the eye?

Line 367

Can you test this with your dataset? Do the datapoints above the dashed line correspond to volcanoes that host a hydrothermal system, whilst those below the line do not?

Line 369

This final statement is not fully clear what you mean, can you expand on this please?

Line 371

Can you show the uncertainties on each bar?

Are bars the best way to show these data, as they are discrete values; what about a scatter plot?

Line 378

What is the dashed line?

Line 388

Concentrations?

Line 389

See previous comments, is convection the only mechanism that can explain the high SO₂ fluxes in some examples? Can you discount degassing prior to MI entrapment?

Line 394

By "loss" do you mean loss from the reservoir? Because the melts will still be degassing by second boiling as they cool and crystallise...

Line 395

I like that you finish on a call for improved integration of petrology and surface gas measurements, I fully support this. However, this statement is currently rather vague and would have more impact if you could tighten this up, perhaps with specific examples of potential hypotheses to test?

Line 403

Consider outlining the basic principle of how this is determined... i.e. based on absorption of UV by SO₂

Line 460

Is this uncertainty incorporated into the errors you report on each flux measurement?

Modest volcanic SO₂ emissions from the Indonesian archipelago

Bani et al

Respond to review

Reviewer 1 :

General comments

Question 1 (Q1) : Extrapolation of sporadic measurements: Most of the estimates for passive degassing are based on measurements performed over very short periods. I would welcome a discussion of the uncertainties in how representative such measurements are likely to be at any given volcano. Some volcanoes exhibit cyclic behavior in their emissions for example (Semeru, Sinabung, Merapi) on timescales from hours to weeks. Have these variations been accounted for? If so how? And how would it impact the reported average emission rate?

Response 1 (R1):

Extrapolation of sporadic measurements is common in volcanic outgassing studies, especially on difficult-to-access volcanoes. In this approach, measurements are made during the passive outgassing phase, which is considered to be more representative of a volcano's state of activity, in contrast to eruptive outgassing, which usually lasts only a short time. There are no continuous and/or systematic measurements of volcanic degassing in Indonesia to identify and quantify any cyclic behavior. The following sentence is added (L291-292): "**We consider this a representative figure, acknowledging that it is based on very limited temporal sampling of the volcanoes in question**".

Q2: Estimates of explosive contributions: The use of the formula relating VEI to SO₂ emissions from (Carn et al. 2016) presents a few difficulties, which I feel have not been addressed in the discussion, and the associated uncertainties have not been stated: 1. The formula was derived from satellite observations. Such observations are unlikely to have included many of the VEI <3 eruptions so common in Indonesia. This is because of several reasons: 1) the small magnitude of the eruptions, 2) the plume altitude often within the troposphere, and 3) the location of the eruptive centers close to the equator, all of which are factors making the satellite retrieval method more difficult. The authors are using a formula which was derived from data specifically excluding the type of eruptions they are trying to assess. 2. There is considerable spread in the data used to derive the formula. The classification of each event into a discrete VEI (integer class, not continuous), and the large range of observed emission rates within each VEI category introduces large uncertainties when using the formula. The reported errors in Table 5 seem to be related to the statistical averaging of multiple years, and do not include what is likely a +/- 40% error from simply using the formula. Although the focus of the paper is clearly on passive emissions, it may be helpful to include a few direct comparisons between documented satellite retrievals over the period 2012-2019, and the approximations taken from the formula. This would provide confidence that the formula is a reasonable approximation, and that the explosive contributions can indeed be estimated in this way.

R2: To improve the assessment of SO₂ from eruptive activities, the satellite data (mainly OMI) available online (<https://so2.gsfc.nasa.gov/>) for the entire Indonesia archipelago over the period 2010-2020 are analyzed and compared to the SO₂ budget from the VEI-SO₂ formula. This additional work shows comparable results for major eruptive discharges but strong disparity remains for minor to moderate eruptions (VEI<3). Out of 110 recorded eruption, 71 were captured by the satellite. Satellite values are systematically lower than the calculated VEI-SO₂ values (new Table 4, new Fig.3). Based on these results, a range of minimum and maximum values is proposed for the SO₂ output during eruptive events and corresponds to 0.12-0.54 Tg/yr. **A new paragraph is added, L223-237.**

Q3: The influence of conduit radius: One of the main messages from the manuscript as I understand it is that the vigor of convective processes within the conduit (largely controlled by conduit radius) is likely the most important factor in explaining which volcanoes exhibit the largest emissions of SO₂. Is there any corroborating evidence suggesting that convection is indeed more vigorous (or that conduit radius is larger) at volcanoes where larger SO₂ emissions are detected? For example, heightened volcanic tremor or inflation/deflation cycles?

R2: Unfortunately, there is no other available data that could support our findings. However there are examples that support our hypothesis, including the Satsuma-Iwojima volcano, where a low-density material was evidenced by muon radiography and interpreted as related to the presence of a magma with 60% vesicularity. Given that highly vesiculated magma is not stable because of permeable outgassing, the low-density body at the shallow depth is taken as evidence of

conduit magma convection consisting of ascending vesiculated magma and descending outgassed magma in a conduit from the magma chamber to the near surface (Shinohara et al., 2012). **This reference is added in the manuscript (ref.55).**

Specific comments

Q3: Table 1 is a nice addition, and a helpful compilation of the literature on the subject. But is a bit busy at the moment, and it may be beneficial to streamline it. I would suggest adding designated columns for: 1) the publication date of the study; 2) the time period covered for the yearly estimates; and 3) a succinct descriptor of the methods (i.e. COSPEC vs DOAS vs Petrological estimates, etc.). The reader could use this table much more efficiently when reading the manuscript.

R3: **Table 1** is now condensed. Only the inventories that specify Indonesia's contributions are maintained.

Q4: L77-78: "... which constitute the main volcanic degassing sources in Indonesia (Table 2, Fig.1)" Consider adding a citation or justification for this statement. Measuring degassing rates from fumarolic fields is notoriously difficult given the lack of a point source. Recent efforts with UV cameras have started to place constraints on such emissions (e.g.; Stebel et al. 2014). This kind of estimates probably suffers the most drastically from the lack of measurements mentioned by the authors for all Indonesian volcanoes. I understand the choice not to include them in this study for that reason. If there is evidence to support their relative contribution is indeed negligible, please include it here.

R4: From several Type B and Type C volcanoes visited in this work, most of them don't have any visible gas emissions. However a through evaluation maybe necessary. The statement is modified as follow (L79-80): **"We focus our efforts on the subaerial type-A volcanoes, which we consider based on field observation to be the main volcanic degassing sources in Indonesia."**

Q5: Figures 1-3 could be combined into one figure. Perhaps even leaving three panels within it for clarity. But most of the information contained in each of these figures is redundant to some extent.

R5: Figure 1, 2 and 3 are grouped into a unique **new Figure 1** as suggested.

Q6: Table 3: When a volcano is labeled as "Quiescent, no gas" or as having "Negligible degassing", no measurement method is associated. Have any (obviously unsuccessful) measurements been performed to determine the lack of emissions? If so, it would be helpful to mention it. If not, what is this determination based on (gas sensors, lack of reports of sulfur smell by nearby communities)?

R6: The terms "Quiescent, no gas" indicate that no gas emission was observed in the main crater or on the flank. To avoid confusion these terms are replaced by **"no activity"**. The "Negligible degassing" correspond to smaller size degassing sites that are difficult to quantify with DOAS and UV-Cam at a distance of few hundred meters. We add the following statement for more clarity (L81-83): **"We use the term 'passive' to refer to the style of gas emission so as to distinguish it from larger, sporadic explosive emissions, though the term can encompass a wide range of sources from magmatic to fumarolic."**

Q7: Figure 4: uses tonnes/day as opposed to Tg/yr. Is there a reason to use one unit over the other? I understand some measurements campaigns only consisted of one day, but it would be clearer for the reader if all values were reported as the extrapolated averaged values with a consistent unit throughout the manuscript (either t/d of Tg/yr).

R7: Unit is converted to **Tg/yr** is the new **Fig.2**.

Reviewer 2 :

Q1: One of the main issues relates to the magnitude of the newly reported SO₂ emissions in comparison to previous satellite measurements, and whether the Indonesian volcanic SO₂ emissions are in fact as 'modest' as reported here. Several of the 47 volcanoes with new SO₂ flux data have SO₂ emissions previously estimated from satellite measurements, and there are some notable discrepancies which could be discussed in more detail. For example, the reported SO₂ fluxes for Kerinci, Ebulo, Lewotolo, Sirung and Slamet are significantly lower (an order of magnitude in some cases) than

space-based estimates. The paper reports SO₂ emissions for Slamet measured in 1991 (~30 tons/day), whereas there are more recent satellite data indicating higher SO₂ fluxes (average of ~200 tons/day). Where both satellite and ground-based data are available at a volcano, the approach here is to use the latter, and hence the implication is that the satellite measurements overestimate SO₂ fluxes, whereas the converse is generally regarded to be true (i.e., satellites underestimate SO₂ fluxes due to limited sensitivity). The authors do point out that the higher satellite-based SO₂ fluxes may be due to the inclusion of eruptive emissions, but this seems irrelevant if the goal is to quantify total volcanic SO₂ output. It would be nice to see more effort to reconcile the ground-based and satellite measurements. This is also relevant to the comparison with other arcs (Figure 6), since the SO₂ data for all the other arcs are also derived predominantly from satellite measurements. If the same reasoning is applied to other arcs (i.e., satellite measurements overestimate the fluxes) then perhaps SO₂ emissions from other arcs are also lower than shown in Fig. 6? This might have implications for the overall conclusions of the study. The analysis is also quite sensitive to the arc lengths, which should be provided.

R1: Additional work was carried out to reconcile ground-based and satellite recordings.

- Satellite data available online (<https://so2.gsfc.nasa.gov/>) for the Indonesian archipelago over the period 2010-2020 are analyzed. Out of 110 recorded eruptions, 71 were captured by the satellite. The SO₂ mass recorded corresponds to 0.22 Tg/yr, which is lower than the figure obtained by the VEI-SO₂ equation. Comparable results are obtained for large eruptions (Merapi, Kelut) (new Table 4, new Fig.3). **A new paragraph is introduced (L223-237) presenting the results obtained using the online satellite data.** The new result corresponds to 0.12-0.54 Tg/yr – a range that integrates both SO₂ from satellite data and from the VEIs. The higher range is equivalent to the previous estimate.

- **The old SO₂ value for Slamet (30 t) obtained in 1991 is now replaced by the recent value of 206 t obtained by satellite.** The new Table 3 also includes the Batu Tara emission (102 t) obtained from the difference between the satellite result (Carn et al., 2017) and the ground result for Lewotolo (this work). **These new entries change the total emission budget for passive degassing from 1.05 Tg/year to 1.15 Tg/year, which does not change the fundamental results and discussion in the manuscript.**

Q2: A related issue is the timescale of the measurements. This is a common (and often unavoidable) issue with ground-based volcanic gas data which are typically collected during brief measurement campaigns. Many of the volcanic SO₂ emissions reported here represent a single day (or a few hours) of measurements, and it raises the question of how representative these are of long-term degassing rates, given that one of the overall conclusions of the paper relates to long-term gas accumulation. On the other hand, some of the satellite measurements span more than a decade (since 2005) and hence arguably provide greater insight into temporal variations in degassing, albeit still on a relatively short timescale. I wondered if there were any other data (e.g., seismic data, heat flux) available from the volcanoes studied here that might be useful for analyzing temporal variations in activity. There is certainly scope to extend the study beyond ~2010-2020. Satellite measurements are available since 2005 for passive and eruptive volcanic SO₂ emissions and since 1978 for eruptive emissions. The latter includes significant eruptions of Indonesian volcanoes such as Galunggung, Colo, Makian [Kie Besi] and Banda Api in the 1980s-90s (notably, these volcanoes currently appear to have negligible SO₂ emissions). Some of these data might further support the paper's conclusions about gas accumulation.

R2: As already emphasized in response to Reviewer 1, extrapolation of sporadic measurements is common in volcanic outgassing studies, especially on difficult-to-access volcanoes. In such an approach, measurements are made during the passive outgassing phase, which is considered to be more representative of a volcano's state of activity, in contrast to eruptive outgassing, which usually lasts only a short time.

In this work, the ground measurements cover the period 2011-2019, and thus the satellite observation period is voluntarily restricted to this period as well.

Almost all type A volcanoes are equipped with drum seismographs hence the analysis of the corresponding papers is difficult and not performed here.

Q3: Another question is how scrubbing of SO₂ and emission of other sulfur species (especially H₂S) might affect the Indonesian volcanic sulfur budget. Most Indonesian volcanoes are 'wet' and as the authors point out there are numerous acidic crater lakes. The authors discuss scrubbing and the total sulfur content of some Indonesian volcanic gases on page 22, but the compositional data (Fig. 7) are limited and do not include many of the major volcanic sulfur sources. If possible,

I think an effort to constrain volcanic H₂S emissions in Indonesia would be valuable. The authors acknowledge the importance of scrubbing or hydrothermal sequestration (e.g., L314) but seem to reject it as a major influence on arc-scale SO₂ emissions. Variable degrees of scrubbing of volcanic SO₂ in ‘wet’ vs ‘dry’ arcs (e.g., much of South America in the latter case) could perhaps partly explain the data shown in Figure 6.

R2: The hydrothermal sequestration certainly plays a role in the total SO₂ degassing budget and is not denied here. However, if the SO₂ scrubbing by the hydrothermal processes was the dominant process, one would expect to see gas composition plotting toward the high X_S_{total} and low SO₂ flux (new Fig. 5). This is the case for Papandayan, Awu, Wurlali, Gamkonora, or Tangkuban Parahu. Other volcanoes, including Lewotolo, Gamalama, Merapi, Sirung, Bromo, Kawah Ijen, Soputan, or Dukono are less influenced by the hydrothermal system. Hence the hydrothermal sequestration alone cannot be responsible for this low SO₂ degassing budget across the Indonesian archipelago.

Q4: The petrological data that underpin some of the main conclusions (Fig. 8, 9) are somewhat limited (as the authors acknowledge) and melt inclusions (MI) are not necessarily a robust indication of primary melt sulfur contents, as they can be affected by various processes (pre-entrapment degassing into a magmatic volatile phase, post-entrapment modification etc.). These issues should at least be acknowledged. Furthermore, there is no discussion of how the other more abundant volatile species (H₂O, CO₂) could play a role in the volcanic processes – the emphasis here is on SO₂ but the other volatiles play a much more significant role in driving volcanic activity.

R4: The following sentence is added in consideration of the MI limits (L379-380): **“assuming that primary sulfur content in the melt inclusions is not affected by post entrapment modification”**.

Unfortunately, only SO₂ is measured here, so it is difficult to draw conclusions about the role of other gases on the behavior of volcanoes in Indonesia. The discussion focuses on how volcanoes with low SO₂ levels are capable of producing eruptions that affect climate? Knowing further that sulfate aerosols are key players in cooling the earth's surface following large eruptions. There has to be a sufficient accumulation of sulfur and of course other gases in the reservoir.

Further comments:

L35: I think ‘arguably’ can be deleted here.

Deleted

L35-47: Perhaps a little odd to have no citations in the first two paragraphs. There are good review articles that could be cited for all the statements given here.

References are added

L46: in terms of satellite measurements, temporal and spatial gaps are much less of an issue now with high-resolution sensors such as Sentinel-5P/TROPOMI. The more significant current challenges are the low sensitivity to weaker volcanic SO₂ sources and the challenge of processing the large volume of data.

Adding in the text the following (L49-50): **“and the challenges of processing large volume of data”**.

L48: the second sentence of this paragraph seems out of place here?

Sentence deleted

L65: Table 1 could perhaps be condensed as I’m not sure all the information is needed – perhaps the more relevant data could be shown here (i.e., the inventories that specifically report data for Indonesia), with a larger compilation in supplementary material? There have certainly been other reviews of the many volcanic SO₂ inventories compiled over the years. Also note that the Carn et al. data cited (last line of Table) spans 2005-2015 (not 2014), but the NASA satellite-based inventory is also updated annually so there are now additional SO₂ emissions data for 2005-2020. Based on recent activity, a couple of new Indonesian SO₂ sources have been added to the database (Arjuno-Welirang and Agung), although this has minor impact on the total Indonesian SO₂ emissions (average ~2.3 Tg/yr).

Table 1 is condensed. Only inventories that specify contributions from Indonesia are maintained (thanks).

L73-75: although the focus is naturally on the Type-A volcanoes, I do wonder about possible H₂S emissions from the numerous Type-B/C volcanoes. Could this be a significant source of sulfur degassing in Indonesia?

The sulfur contribution to the atmosphere from type B-C volcanoes is not investigated here. But most of the type B-C volcanoes we’re encountered in the field (at least to accessible points) are inactive and a few exhibit steaming points. A thorough inventory is needed to address this question which is beyond the scope of this work.

L77: ‘Halmahera’ (typo). Plus, should it be Sulawesi-Sangihe for consistency with Table 2? Halmahera corrected and Sulawesi-Sangihe changed to Sangihe in the Table 2

L92: ‘subaerial’

corrected

L97: Batu Tara is also listed in the satellite-based inventory, though the SO₂ emissions are merged with Lewotolo. Arjuno-Welirang has recently been added to the database too.

The Batu Tara contribution is obtained from the difference between the satellite results (ref 3) and the Lewotolo emission obtained in this work.

L139: Recommend providing the alternate name for Rokatenda (Paluweh) to avoid any confusion.

Paluweh added

L144-146: Based on satellite observations, evidence suggests that SO₂ emissions from Batu Tara must be more significant than Banda Api. There is a substantial SO₂ signal over Batu Tara in some years, but no passive degassing has been detected from space at Banda Api to date.

Batu Tara emission is added (thanks).

L158: Kie Besi more commonly known as Makian?

Added in in brackets

L180: although there is some agreement, these rankings would look somewhat different if the satellite SO₂ inventory for Indonesia (2005-2015 or 2005-2019) was also considered. Summing the average SO₂ fluxes in 2005-2019 for the 19 Indonesian volcanoes with emissions detected from space, the total SO₂ flux is ~6200 Mg/d (~2.3 Tg/yr), i.e., more than twice that reported here.

We added the following statement to account for such a fluctuation (L161-163): **“We emphasise that this figure is representative of the periods of observations and must be viewed cautiously but we believe it gives a useful guide to the scale of emissions at the scale of the entire arc.”**

L207: should it be ref. 3 cited here, rather than ref. 15?

modified to ref.3

L210: it may be partly true that the satellite measurements include more vigorous eruptions, though many of the larger eruptions are filtered out by using a threshold SO₂ column amount in the satellite data analysis. Furthermore, eruption clouds tend to drift away from the volcanic source, whereas the passive emissions are derived only from satellite data collected close to the volcano in question. But regardless of the origin of the SO₂, both passive and eruptive emissions contribute to the overall arc flux, hence I don't see the inclusion of eruptive emissions as an issue here.

The following sentence is deleted to avoid confusion: **“This likely reflects inclusion of more vigorous eruptions in the spaceborne-derived catalogue, which spans a longer time period than our observations.”**

L213-214: without seeing a detailed list of the eruptive events, it is difficult to assess the accuracy of this statement, unless the satellite data were analyzed as part of this work.

The satellite data over the 2010-2020 period are analyzed and the result is added into the manuscript (new Table 4 and new Fig.3).

L225: Note that some of the explosive eruptions (e.g., multiple Sopotan and Sinabung eruptions) were detected in the satellite data and would be listed in the NASA database of eruptive volcanic SO₂ emissions (https://disc.gsfc.nasa.gov/datasets/MSVOLSO2L4_4/summary). Hence, I don't think it is necessary to use the VEI-SO₂ relationship (which has significant uncertainty) for all the explosive eruptions, if actual measurements are available.

Satellite data and VEI-SO₂ results are both taken into account in the corrected version. A range of results is provided based on the combined result.

L259: since arc lengths are a factor in the analysis, the arc lengths used in the paper should be provided. The results in Fig. 6 are quite sensitive to this parameter and more details on how it was measured or obtained (including any uncertainty) would be useful.

We consider that the length of an arc is proportional to the length of its corresponding trench that we retrieve from bathymetry maps. The same approach was used by Hilton et al (2002). We now compare our estimations with that of Hilton et al and subsequently adjust the new fig.4. The value for South America includes Colombia, Ecuador, Peru and Chile. The value for Japan includes the arcs of Japan, Ryukyu and Nankai whilst in Italy it is length of Calabrian arc that is considered. We also added the length in the new Figure 4.

L278-279: though ironically, both Ambrym and Kilauea have had much reduced SO₂ emissions since 2018-19, which shows the dynamic nature of volcanic degassing and the importance of the timescale of observations.

That is true – The new statement (L161-163) also refers to that.

L280-282: the 1982-83 Galunggung (Java) eruption produced ~2 Tg SO₂, so there have also been relatively recent Indonesian eruptions with comparable output to those listed here.

Galunggung 1982-1983 eruption is now mentioned (L302-303).

L355-356: presumably Sopotan's frequent explosive eruptions are linked to its volatile-rich magma (not just sulfur but also H₂O, CO₂).

Given the high sulfur content in the melt inclusion and the basaltic composition, one can assume volatile rich magma on Sopotan. Sopotan was missed out in the old Fig.7 and now corrected in the new Fig.5.

L357: I think it is clear that Dukono is a special case among the currently active Indonesian volcanoes, since its activity (with frequent emission of volcanic ash; presumably juvenile magma) seems to shift between continuous eruption and passive degassing. Perhaps during continuous eruptive activity there is less conduit convection. Given that eruptions usually involve a greater flux of magma, this would certainly explain the high SO₂ flux at Dukono.

There is a continuous erupting activity on Dukono, but with more or less stronger intensity.

L405: although the spectroscopic techniques used to measure the volcanic SO₂ fluxes are described here in general terms, it is a bit difficult to assess the data quality at individual volcanoes without further information on measurement conditions (e.g., were the plumes optically thick/condensed, possible aerosol impacts., etc.)

Ideal measurement conditions on the field are rare but precaution were taken to increase chances of having good data. Both traverse and stationary recording were carried at distance varying between few tens of meters from the craters to around 5 km downwind, depending on the access difficulties, the plume size and the volcanic activity. UV-cam measurements were performed mainly in the late morning before the clouds started to form generally at about 09-11 am. Image calibrations were carried out regularly during measurement to correct for rapid change of light intensity. This information is now added (485-486).

Figure 4-5, 8: these figures could be improved. Bar charts may not be the best way of presenting these data, as there is a lot of white space and the text is small.

Figures are improved and grouped.

Figure 6: A) a log scale might work better here as the Indonesian volcanic contribution is difficult to see. And perhaps also show only those inventories which report data for Indonesian volcanoes?

The log scale is used in the new Fig.4 and only the inventories with Indonesian estimates are kept.

Figure 9: Are the points plotted along the x-axis (i.e., ~zero or very low S in MI) actual measurements?

No, and we add the following in the legend Note (new fig.6): **“we did not find melt inclusion values for data points with zero SO₂ flux and conversely.”**

Reviewer 3 :

- I have made my comments directly on the manuscript file, so please see attached.

Comments and suggestions are taken into account in the corrected manuscript (thanks).

- Uncertainties: In data-rich papers, such as this one, the propagation of uncertainties is really important. Throughout the paper, I would encourage the authors to explain clearly what each reported uncertainty represents (i.e. a measure of variance on repeat measurements, or an absolute uncertainty associated with the measurements themselves, such as wind speed), and how these are carried through to the final arc-scale fluxes. These error bounds should be displayed on figures, even when they are in the form of bar charts. On this note, I might suggest that some of the data might be better represented with scatter plots rather than bars, when illustrating discrete values.

The errors relating to light intensity, wind direction and wind speed are applied to each traverse and profile, then the mean value is calculated for each series of measurement with the corresponding standard deviation. The global estimate for the Indonesian arcs is the sum of the mean values. This information is now added (L496-499). Bar charts are replaced by scatter plots the new Fig.2.

- Figures and tables: There are some repetitions between sequential figures and tables; for example, between figures 1-3 and between tables 3 and 4. In the interests of being concise, but also grouping similar datasets together, I would encourage the authors to think about whether any of these could be combined, either by merging or by creating multi-part (a and b) figures.

Table 3 and Table 4 are merged to a new Table 3 whilst Fig.1, Fig.2, and Fig.3 are merged into the new Fig.1

- The phrasing “sulfur budget” or “degassing budget” is used frequently throughout the paper. To me, a budget requires an evaluation of inputs and outputs... the authors are presenting only outputs, and therefore fluxes. Have a think about whether budget is really the right term here.

Both sulfur budget and SO₂ flux are used in the ms but budget refers more to the total amount of gas released.

- The units reported switch between Tg and Mg, and once kt, throughout the paper. Consider keeping the units consistent for each of comparison.

Units are now limited to Mg and Tg

- Comparing time-averaged vs “instantaneous” measurements. Carn et al 2017 report time-averaged fluxes over a decadal period, whereas the measurements in this study are (I think) predominantly campaign-based? It is an interesting question to what extent long-term averages can be compared to “instantaneous” flux measurements, and this could have been brought into the discussion more strongly as I think it is very relevant to the points being made.

As already stated above, the extrapolation of sporadic measurements is common in volcanic outgassing studies, especially on difficult-to-access volcanoes. In this approach, measurements were made during the passive outgassing phase, which is considered to be more representative of a volcano's state of activity, in contrast to eruptive outgassing that lasts only for a short time. The added following statement (L161-163): **"We emphasise that this figure is representative of the periods of observations and must be viewed cautiously but we believe it gives a useful guide to the scale of emissions at the scale of the entire archipelago."**

- In the discussion, statements are made regarding the link between sediment inputs into the subduction zone and the sulfur content of the magmas that yield the emitted gases. But is the relationship between sediment flux, primary melt S contents, and melt inclusion S contents this simple though? The S in melt inclusions is affected by the sulfide saturation of mantle source, sulfide saturation and sequestration during magma ascent and storage, and importantly fluid exsolution prior to melt inclusion entrapment... can S loss prior to MI entrapment be discounted as a contributing factor to the lower than expected MI concentrations for Dukono, for example? Are there any constraints on entrapment pressures in the original studies where these data are sourced from?

Only a few works have focused on the initial pressure/temperature conditions of the source and the statement L426-429 calls for more petrology works on volcanic products. For the S concentrations in the melt, we used the maxima reported which are assumed to be representative of initial S contents in the melt. We add a statement (L385-386) on the assumption that the melt inclusions are little to no affected by post-trapping modification.

- In the discussion, it would be good to see a slightly more nuanced discussion regarding the interplay between temperature, composition, and redox on sulfide saturation, and thus dissolved sulfur contents in magmas of different compositions. Statements are nuanced as suggested.

- When discussing figure 9, are there other processes, such as gas fluxing of a segregated volatile phase, that could explain high SO₂ fluxes without the need to invoke extensive convection, particularly for more evolved compositions? Where do you envisage this degassed magma accumulates if not erupted?

The accumulation of the degassed magma is a topic that still needs to be better constrained and convection process may play some key roles. But that is beyond the scope of this work.